# Rpn11-mediated ubiquitin processing in an ancestral archaeal ubiquitination system

Adrian C.D. Fuchs[1], Lorena Maldoner[1], Matthias Wojtynek[1], Marcus D. Hartmann [1] & Jörg Martin [1]

While protein ubiquitination was long believed to be a truly eukaryotic feature, recently sequenced genomes revealed complete ubiquitin (Ub) modification operons in archaea. Here, we present the structural and mechanistic characterization of an archaeal Rpn11 deubiquitinase from *Caldiarchaeum subterraneum*, CsRpn11, and its role in the processing of CsUb precursor and ubiquitinated proteins. CsRpn11 activity is affected by the catalytic metal ion type, small molecule inhibitors, sequence characteristics at the cleavage site, and the folding state of CsUb-conjugated proteins. Comparison of CsRpn11 and CsRpn11–CsUb crystal structures reveals a crucial conformational switch in the CsRpn11 Ins-1 site, which positions CsUb for catalysis. The presence of this transition in a primordial soluble Rpn11 thus predates the evolution of eukaryotic Rpn11 immobilized in the proteasomal lid. Complementing phylogenetic studies, which designate CsRpn11 and CsUb as close homologs of the respective eukaryotic proteins, our results provide experimental support for an archaeal origin of protein ubiquitination.

[1] Department of Protein Evolution, Max Planck Institute for Developmental Biology, Max-Planck-Ring 5, 72076 Tübingen, Germany. Correspondence and requests for materials should be addressed to J.M. (email: joerg.martin@tuebingen.mpg.de)

Protein ubiquitination regulates a wide range of physiological processes, often by targeting modified substrate proteins for degradation by the proteasome[1]. Substrates can be (poly) ubiquitinated via isopeptide bonds by a cascade of E1, E2, and E3 enzymes, a process that is reversed by specific deubiquitinases, among them zinc-dependent JAMM (JAB1/MPN/Mov34) proteases like the proteasomal Rpn11[2–4]. Although classical ubiquitination is thought to be restricted to eukaryotes, several distinct but evolutionary-related pathways are present in prokaryotes, which usually comprise just one ubiquitin (Ub)-, E1- and JAMM homolog each[5]. These systems are typically involved in cofactor synthesis, such as thiamine or molybdenum cofactor, but can also be bifunctional, as shown for the archaeon *Haloferax volcanii*. Here, SAMPs (small archaeal modifier protein) are also used to modify protein functions or to mark substrates for proteasomal degradation[6,7]. However, given the absence of E2 and E3 enzymes in *H. volcanii* and other SAMP-containing archaea, and the evolutionary distance between SAMP and Ub, it seems unlikely that the SAMP system is the direct progenitor of the eukaryotic ubiquitination system[8].

The first complete archaeal Ub modifier system, including E2 and E3 enzymes, was discovered in 2011 in *Candidatus Caldiarchaeum subterraneum*[9]. Remarkably, all involved proteins were found to be far more similar to their eukaryotic counterparts than to any prokaryotic homologs. Thus, eukaryote-type protein ubiquitination might already have evolved in archaea. This interpretation was supported by the more recent discovery of Asgard archaea, which harbor several ancestral eukaryotic features and whose genomes also contain potential Ub modifier systems[10]. In line with this, the capability of *C. subterraneum* E1, E2, and E3 enzymes to conjugate Ub to an artificial substrate protein suggests that these primordial modification systems may have an equivalent function in these archaea[11].

*C. subterraneum* Ub (CsUb), E1, E2, and E3 are organized in a single operon, adjacent to a second E3 protein and a JAMM protease (CsRpn11) with high sequence similarity to eukaryotic Rpn11[9]. In eukaryotes, this deubiquitinase cannot act on its own, but is embedded within the proteasome lid, wherein it forms a heterodimer with an inactive homolog, Rpn8[12]. This organization is retained in paralogous eukaryotic structures, known as PCI complexes (Proteasome—COP9 signalosome—Initiation factor 3), but not known to exist in archaea[13,14]. In contrast to Rpn11-related proteases like AMSH, which act on Ub in isolation and are constitutively active[15–17], eukaryotic Rpn11 was shown to undergo a conformational switch at its Ins-1 site upon Ub binding. This was interpreted to be a regulatory feature, allowing deubiquitination to occur only after the Ub-conjugated substrate became committed to unfolding and degradation by the proteasome[18].

Here, we analyze the evolutionary relationships of CsRpn11 and CsUb with their eukaryotic homologs. Our bioinformatic, structural, and biochemical results reveal unique features of the archaeal proteins, but also remarkable conceptual similarities with their eukaryotic counterparts. They not only bolster the hypothesis of an ancestral archaeal Ub modification system, but also provide insight into the functioning and mechanism of eukaryotic Rpn11-like enzymes.

## Results and discussion
### Co-evolution of archaeal ubiquitins and Rpn11-like proteins.
To put the archaeal Ub modification system into an evolutionary context, we started out our investigation with an in-depth sequence analysis of CsRpn11 and CsUb. While CsRpn11 can be easily recognized as a member of the JAMM protease family, its role in a putative ancestral ubiquitination system should be reflected by its particular evolutionary relationship to eukaryotic Rpn11, as compared to other archaeal JAMM proteases. To look into this aspect, we built a bioinformatic map of the JAMM protease universe, by searching for JAMM homologs in the non-redundant protein sequence database and clustering them by pairwise sequence similarity (Fig. 1a). In the resulting CLANS map[19], the eukaryotic active PCI complex deubiquitinases (CSN5, Rpn11; eIF3H is inactive) form a single cluster that is centrally located between the other eukaryotic clusters and forms the strongest connections to prokaryotic sequences. Further removed are the inactive PCI complex subunits CSN6, eIF3F, and Rpn8 that also cluster in close vicinity to one another, highlighting the paralogous relationship between Rpn8–Rpn11, eIF3H-eIF3F, and CSN5-CSN6.

In contrast to these well-characterized proteins, many prokaryotic JAMMs are of unknown function. Recently, two groups have been described[20]: JAMM-2 lacks independent activity, whereas isolated JAMM-1 can act as desampylase. While the clusters formed by these sequences lie distant from the eukaryotic ones, our analysis reveals a novel third group, JAMM-3, with higher similarity to eukaryotic Rpn11 and CSN5 but only poor connections to other eukaryotic family members. The closest connections to Rpn11, however, are made by JAMMs from *C. subterraneum* and Asgard archaea, which are centrally located between the Rpn11, Brcc36, and AMSH clusters[21]. Remarkably, while prokaryotic JAMMs which cluster distant from Rpn11 are functionally associated with just Ub and E1 homologs, sequences that cluster closer to Rpn11 are found in operons with Ub-, E1-, and E2-homologs. Finally, *C. subterraneum* and Asgard archaea, whose JAMM sequences are most similar to eukaryotic Rpn11, contain in addition E3-like RING finger proteins, further complementing the ubiquitination machinery. Thus, it appears that the increase in complexity of the Ub modification system was accompanied by the evolution of increasingly Rpn11-like JAMM sequences, with CsRpn11 at the apex of this process.

Likewise, a cluster analysis of Ub-like proteins (Fig. 1b) reveals eukaryote-like Ub sequences in E2- and E3-containing archaea, while other prokaryotes only encode for more distant homologs. Urm1, previously proposed as potential Ub ancestor[3], is found distant from classical Ub and more closely related to prokaryotic homologs like MoaD and ThiS. Again, the closest connections to eukaryotic ubiquitin are made by sequences from Asgard archaea and *Caldiarchaeum* (dark violet, magenta; enlarged inset), CsUb being up close with 31% sequence identity (Fig. 2a). The Ub modification system in these organisms thus has the features expected for a predecessor of the eukaryotic Ub modification system.

**Crystal structure of the *C. subterraneum* ubiquitin CsUb**. To learn more about the function and mechanism of the individual components comprising the archaeal Ub modification system, we set out to characterize them in molecular detail. As *C. subterraneum* cannot be cultured, we synthesized CsUb (*Csub_C1474*) and CsRpn11 (*Csub_C1473*) genes and recombinantly produced the proteins in *E. coli*. Fitting to the thermophilic nature of *C. subterraneum*[9], CsUb is an extremely thermostable monomeric protein (Tm > 95 °C in circular dichroism heat-denaturation studies, Supplementary Fig. 1). The CsUb gene encodes for a precursor of 87 amino acids (Ub-pre). After cleavage by CsRpn11 (see below), the carboxyl group of Gly78, which is the reactive part in activation and conjugation reactions, is exposed in mature CsUb[11]. While an unpublished CsUb NMR solution structure has been deposited (PDB 2MQJ), in crystallization trials with mature CsUb we obtained diffracting crystals

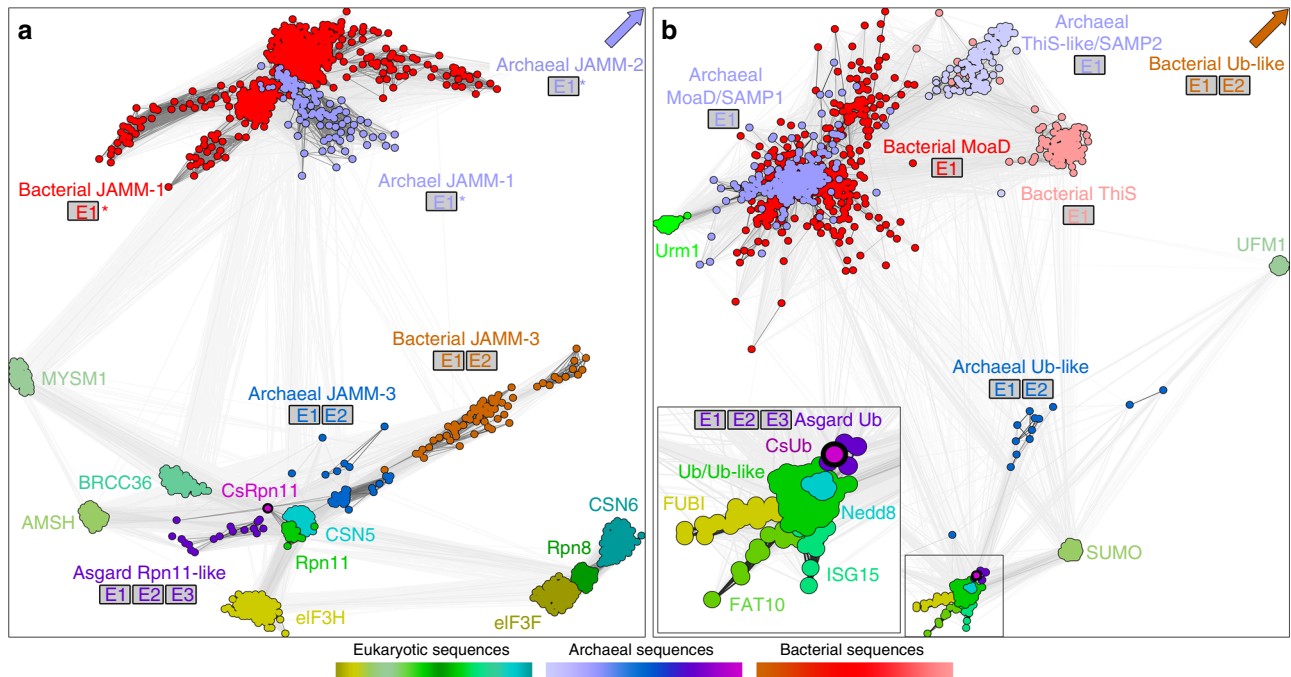

**Fig. 1** Sequence relationships of JAMM proteases and ubiquitin-like proteins. **a** A cluster map of 1543 JAMM core domain sequences. Sequences are represented by dots, the line coloring reflects BLAST *p* values; the darker a line, the lower the *p* value. For simplicity, the map does not include eukaryotic sequences of the prp8, KIAA0157, and FLJ14981 type as well as bacterial lysM-associated JAMM sequences. For prokaryotic JAMMs, the genomic or functional (asterisks) association with E1/E2/E3 proteins is shown. MYSM1 is a histone H2A deubiquitinase, BRCC36 and AMSH are K63-specific deubiquitinases, and CSN5 cleaves Ub-like Nedd8[59]. CsRpn11 is marked in magenta. **b** A cluster map of 1473 Ub-like sequences, prepared in the same way as the JAMM cluster map. UFM, Urm1, Sumo, Nedd8, ISG15, FAT10, and FUBI are eukaryotic Ub-like modifiers that can be conjugated to target proteins in a similar fashion as Ub[8]. MoaD and ThiS-like proteins function in prokaryotic cofactor synthesis, but can also modify target proteins in some archaea[6]. The inset shows the ubiquitin cluster containing CsUb, marked in magenta

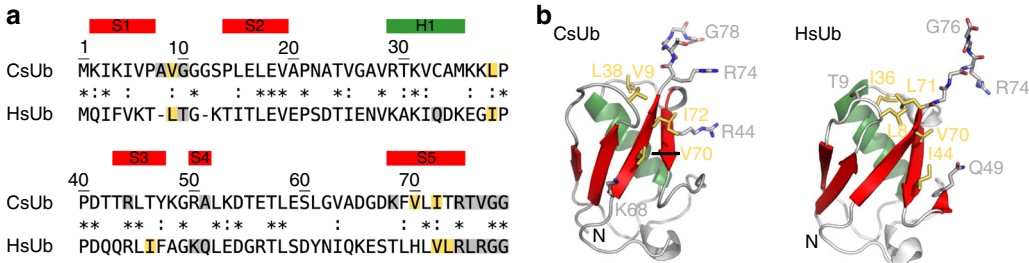

**Fig. 2** Crystal structure of the ubiquitin CsUb from the archaeon *C. subterraneum*. Sequence alignment (**a**) and structure comparison (**b**) of CsUb and *H. sapiens* Ub (HsUb; PDB 1D3Z). The five-stranded sheet (red) and the central helix (green) are highlighted. Residues that form important backbone (gray) or hydrophobic (yellow) interactions with CsRpn11 or AMSH are shown as sticks (see Fig. 3). In the alignment, identical (asterisks) and similar (colons) residues are indicated

that yielded data to 1.05 Å, enabling structure analysis at atomic resolution (Fig. 2b). We could solve the structure via molecular replacement using human ubiquitin as a search model, revealing for CsUb the classical β-grasp fold with a two-residue insertion in the first-loop region. In agreement with the sequence similarities in the cluster map (Fig. 1b), CsUb also superimposes best with human Ub (DALI Z-Score 12.2), but is more distinct from so far characterized prokaryotic Ub-like proteins, such as *H. volcanii* MoaD/SAMP1 (Z = 6.9) or ThiS-like/SAMP2 (Z = 5.4; Supplementary Table 1). Notable structure features are a hydrophobic patch (Fig. 2b, residues Val9, Leu38, Val70, and Ile72), which in eukaryotic Ub is known to be critical for interactions with other proteins[22], and eight surface-exposed Lys residues. In eukaryotes, the side chains of Lys residues are used for formation of isopeptide-linked poly-Ub chains, resulting in a variety of linkage types[23]. Though the CsUb Lys are found at different surface

locations, except for Lys31 (Lys29 in HsUb; Supplementary Fig. 2), a similar formation of poly-CsUb chains cannot be excluded[11], given that eukaryotic Ub homologs, such as Sumo or Nedd8, also use differently located Lys for such assemblies[24,25].

**Structure of the *C. subterraneum* deubiquitinase CsRpn11.** To determine crystal structures for CsRpn11, we used the full-length wild-type form and a series of truncated proteins with deletions C-terminal of the JAMM core domain. Using these constructs, we obtained crystals for full-length CsRpn11 and truncated CsRpn11[Δ149–202], for which we collected data to 1.65 Å and 1.35 Å resolution. The CsRpn11 structure could be solved by molecular replacement using the Rpn11 coordinates from a *Saccharomyces cerevisiae* Rpn8–Rpn11 heterodimer. We located two protomers in the asymmetric unit, which form a homodimer grossly reminiscent of the Rpn8–Rpn11 heterodimer. However,

while the latter is assembled in an antiparallel fashion, the CsRpn11 homodimer is parallel, with an interface formed by the first and the fourth alpha-helix, H1 and H4, of CsRpn11 (Supplementary Fig. 3). In the truncated CsRpn11$^{\Delta 149-202}$ structure, however, which is lacking H4, and which we solved on the basis of the full-length Rpn11 coordinates, this dimerization is not observed. Notably, based on static light-scattering (SEC-MALS) measurements, CsRpn11 appears to be monomeric in solution (Supplementary Fig. 4). Such a disparity, dimeric crystal forms vs. monomers in solution, was already noted for other JAMM and Rpn11-like proteins[16,26,27]. The physiological relevance of these dimers is unclear, considering that the dimerization interfaces in the respective crystal structures differ (Supplementary Fig. 3). Moreover, JAMM protease from *Pyrococcus furiosus* has been found to be a monomer/dimer mixture in solution, of which both forms are active[20].

In structure superimpositions, CsRpn11 displays the highest similarity to *S. cerevisiae* Rpn11 (DALI Z-Score 21.2), but also superimposes well with CSN5 ($Z = 19.2$) or BRCC36 ($Z = 19.7$). By contrast, it is less similar to AMSH ($Z = 15.3$) or known structures of other prokaryotic JAMMs, such as *P. furiosus* JAMM-1[20] ($Z = 15.8$; Supplementary Table 2). Common to all these proteins is the JAMM protease core, a seven-stranded β-barrel with the topology S1-S3-S2-S4-S5-S6-S7-S1, that is surrounded by three helices (Fig. 3a, b). This region is highly conserved between CsRpn11 and its eukaryotic homologs (33% identity to human HsRpn11) and contains the conserved active site residues Glu24, His83, His85, Ser93, and Asp96 (marked by black boxes in Fig. 3a). The core is flanked by helices H4 and H5, which are not required for CsRpn11 activity (see below), and which are absent in SAMP-specific JAMM proteases.

When comparing CsRpn11 with eukaryotic Rpn11, an obvious difference concerns the positioning of the latter protein within the proteasomal lid. Eukaryotic Rpn11 has an extra N-terminal helix (residues 1–23 in ScRpn11), further C-terminal helices, and an insertion between S6 and S7, known as Ins-2[17] (Fig. 3a, b; N and C termini are not resolved). These elements anchor the protein in the proteasomal lid complex[12,28], but are not involved in substrate binding. By contrast, in the non-proteasomal JAMM-type deubiquitinase AMSH, Ins-2 interacts with the proximal Ub in di-Ub substrates and determines Ub–Ub linkage-type specificity[15]. The absence of Ins-2 in CsRpn11 suggests that the archaeal enzyme either does not encounter poly-CsUb chains, or—should they occur—has no preference for certain linkage types.

Another obvious difference concerns the conserved Ins-1 region, an insertion in the JAMM core, which recognizes and positions the Ub C terminus for cleavage. In eukaryotic unliganded Rpn11, Ins-1 is found in a single, well-defined closed conformation, blocking access to the catalytic groove (Fig. 3b and Supplementary Fig. 5)[17,28]. In CsRpn11, however, Ins-1 displays a certain degree of conformational flexibility: In one protomer of the full-length structure it is fully ordered, but partially disordered in the other. In both, the conformation is different from the eukaryotic one, but both obstruct the catalytic groove in a similar fashion, suggesting a conserved function of this insertion (Fig. 3b and Supplementary Fig. 5).

**CsUb induces a conformational switch in CsRpn11 Ins-1**. To obtain structural insight into the CsRpn11–CsUb interaction, we performed co-crystallization trials of CsRpn11$^{\Delta 149-202}$ and CsUb and obtained crystals diffracting to 1.85 Å resolution. The structure was solved by molecular replacement using the CsRpn11$^{\Delta 149-202}$ and CsUb coordinates and indeed revealed

the architecture of the CsRpn11–CsUb complex (Fig. 3b and Supplementary Figs. 5 and 6). The complex is stabilized by a network of hydrogen bonds and a hydrophobic interaction involving the CsUb hydrophobic patch (Figs. 2b and 3e). Interestingly, the CsUb C-terminal tail is now found in a more elongated conformation, extending into the CsRpn11$^{\Delta 149-202}$ binding groove which is bounded by H3 on one, and the Ins-1 region and H2 on the other side (Fig. 3b and Supplementary Figs. 5 and 6). Compared to unliganded CsRpn11, we see that Ins-1 has undergone a major transition in the complex with CsUb. Via an induced-fit-binding mechanism it is now ordered into a β-hairpin that associates with a newly formed β-strand of the CsUb C terminus. The result is a three-stranded β-sheet, which positions CsUb correctly in the binding groove and its ultimate residue Gly78 precisely at the cleavage site (Fig. 3b–d). In support of this positioning function, mutations in eukaryotic Rpn11 Ins-1 lead to severe catalytic defects[17]. Moreover, in structures of the eukaryotic Rpn11-Ub and AMSH-Ub complexes, but also unliganded AMSH, Ins-1 forms a short β-hairpin and the active site is accessible[13,15,28]. It has been proposed[18] that the closed Rpn11 Ins-1 conformation is pivotal to prevent premature Ub cleavage at the proteasome, with Rpn11 becoming only active through the mechanical pulling on the substrate by the 19S AAA motor[18]. A flexible residue in the Ins-1 loop (Gly77 in ScRpn11; highlighted in magenta in Fig. 3a) would give it the necessary malleability to adopt the closed conformation, whereas constitutively active DUBs, like AMSH, which function without the intricacies of proteasomal coupling of unfolding and degradation, have a more rigid Pro at the equivalent position and constitutively feature a β-hairpin also in the Ub-free state[15]. CsRpn11, however, contains a Pro in the Ins-1 loop, and yet Ins-1 exists in two states and readily undergoes a conformational switch upon substrate binding, as seen in our structures above (Supplementary Fig. 5). The ability to undergo this transition thus cannot solely hinge on the presence of a flexible Gly residue in Ins-1. Moreover, the transition is not restricted to a proteasome-embedded Rpn11, but an integral feature also of CsRpn11-like proteins acting in solution. Our results show that the Ub-binding mode of AMSH, Rpn11, and CSN5, which is similar to that of the SAMP–JAMM interaction[20], was already established in an archaeal ancestor, and that CsUb binding alone can trigger the conformational switch in Ins-1, obviating the need for additional factors.

The CsRpn11–CsUb complex structure described above represents the state immediately after the cleavage reaction, with the CsUb C terminus coordinating the catalytic zinc ion. This ion, however, is found with only partial occupancy in our structure. In an attempt to obtain full occupancy, we soaked CsRpn11$^{\Delta 149-202}$–CsUb crystals with ZnCl$_2$ and indeed obtained the structure of a completely Zn$^{2+}$-bound complex. Unexpectedly though, this structure now represents a later step in the catalytic cycle: In addition to Zn$^{2+}$, a new catalytic water molecule is bound (Fig. 3c). It is found in the same place as in the unliganded CsRpn11 structure, but in the complex it displaces the CsUb C terminus into the second coordination sphere of the metal, which can be regarded as the initiation of CsUb release.

**Substrate discrimination by CsRpn11 and CsJAMM protease**. Eukaryotic Ub is synthesized in precursor form and processed by various deubiquitinases (DUBs) in the cytosol[29]. In contrast, the embedding of eukaryotic Rpn11 in the proteasomal lid precludes this physiological activity and limits its action to poly-Ub proteins, which are targeted to the proteasome for degradation. This constraint does not apply to the archaeal CsRpn11,

which can act on both CsUb-pre and Ub conjugates[11]. In fact, homologs of eukaryotic non-JAMM DUBs have not been found yet in archaea. We monitored CsUb precursor (CsUb-pre) cleavage by CsRpn11 via SDS-PAGE and mass spectrometry (Fig. 4a, c). The determined masses show that CsUb-pre is cleaved precisely after the C-terminal Gly–Gly motif (Fig. 4a), while the *C. subterraneum* SAMP precursors, Csub_C0702 and Csub_C1603, were not processed at all (Fig. 4e). Likewise, neither linearly linked human Ub (HsUb$_2$), nor Lys48- or Lys63-isopeptide linked HsUb$_2$ were suitable substrates (Fig. 4e). In

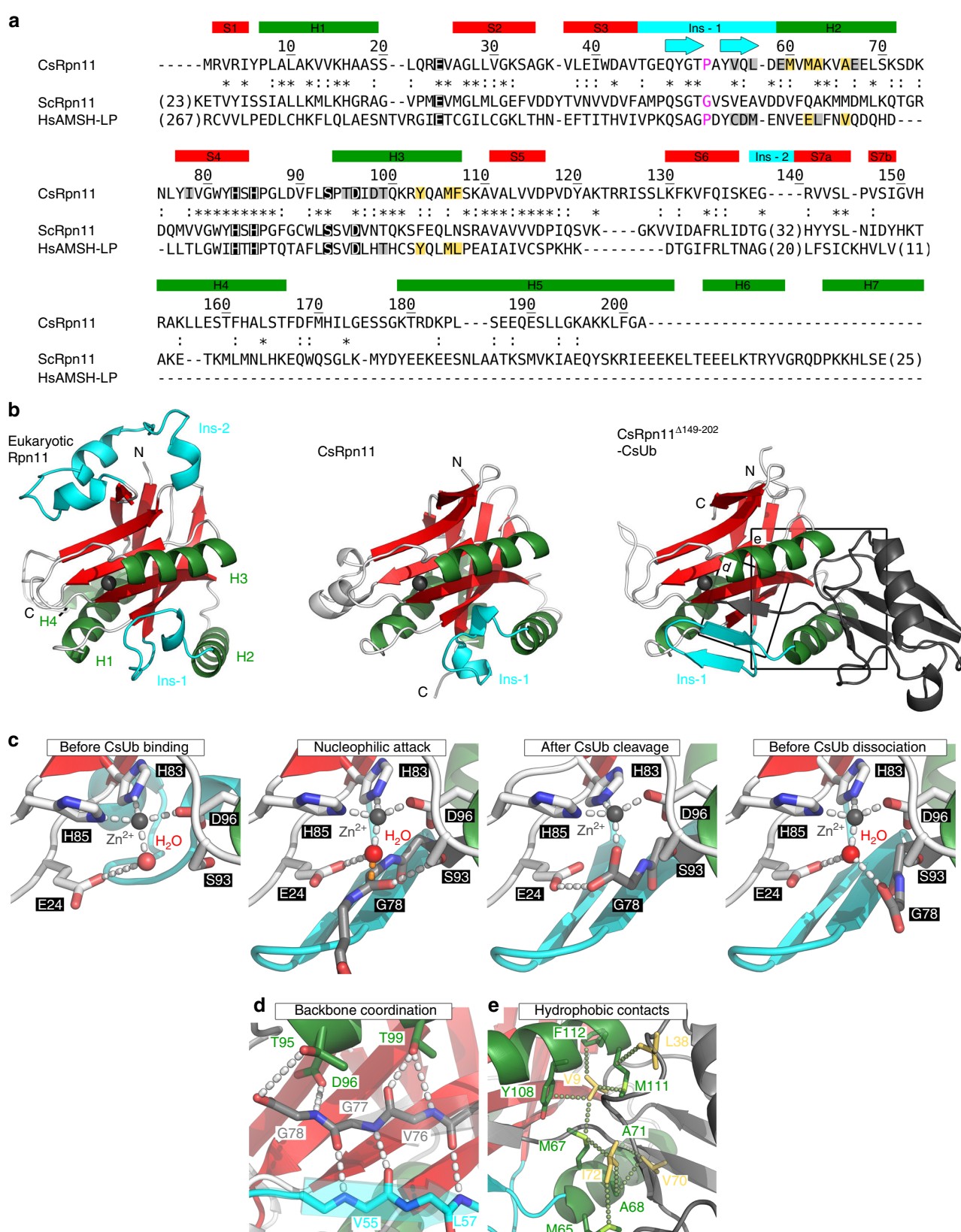

this respect, CsRpn11 markedly differs from the rather promiscuous *P. furiosus* SAMP-protease JAMM-1, which readily cleaves different *H. volcanii* SAMPs and even human Ub conjugates[20]. The high specificity of CsRpn11 for CsUb may reflect its co-existence with a CsSAMP/CsJAMM system in *C. subterraneum* and the resulting need to avoid cross-reactivity. In support of this hypothesis, we find that CsJAMM can only cleave CsSAMP precursors, but not CsUb-pre or HsUb₂ (Supplementary Fig. 8a). In contrast, less discriminative JAMM proteases from other archaea may not be under such constraints, as in vivo they solely encounter their respective SAMP proteins as substrates.

The kinetic parameters of the CsRpn11 enzyme were determined for full-length CsRpn11 and CsRpn11$^{\Delta 149-202}$ via gel band quantification and microscale thermophoresis (MST; Fig. 4b, c). At 45 °C, truncated CsRpn11 processed CsUb-pre with a $K_M$ of 24.2 μM and a $k_{cat}$ of 3.5 min⁻¹, and bound the reaction product with a $K_D$ of 14.6 μM. With a $K_M$ of 31.6 μM and a $k_{cat}$ of 4.0 min⁻¹, the full-length version displayed similar reaction parameters, indicating that the CsRpn11 C-terminal helices are indeed not critically required for catalysis. Though for eukaryotic Rpn11 the reaction environment is different within the assembled proteasome[12], an isolated Rpn8–Rpn11 complex processes the small model substrate Ub-TAMRA at similar rates ($K_M = 20$ μM; $k_{cat} = 0.95$ min⁻¹ [17]), while AMSH cleaves its preferred substrate, Lys63-isopeptide linked diubiquitin, more effectively ($K_M = 18.8$ μM; $k_{Cat} = 90$ min⁻¹ [16]). We note, however, that *C. subterraneum* is a thermophile[9], implying a faster substrate turnover for CsRpn11 under physiological growth conditions (Supplementary Fig. 9).

**CsRpn11 activity hinges on the catalytic metal ion type**. In their catalytic cycle, JAMM proteases use Zn²⁺ and a Glu residue to polarize a water molecule, enabling it to perform a nucleophilic attack at the amide bond following the C-terminal Ub Gly–Gly motif[15]. With the crystal structures shown here, we cover three stages of the reaction cycle: In the CsRpn11 structure, the deubiquitinase is armed for cleavage, while the CsRpn11$^{\Delta 149-202}$–CsUb complex represents the state immediately after the cleavage reaction (Fig. 3c). The structure of a completely Zn²⁺-bound CsRpn11$^{\Delta 149-202}$–CsUb complex depicts the next step in the catalytic cycle, where a newly bound catalytic water molecule displaces the CsUb C terminus to initiate substrate release. As these results suggested a highly specific role for Zn²⁺, we assessed the influence of various active site metal ions on the catalytic properties of CsRpn11. As expected, CsRpn11$^{\Delta 149-202}$ was inactive toward CsUb-pre when its active site zinc ion was stripped with EDTA (Fig. 5a). However, complete activity could be restored upon re-addition of Zn²⁺, Co²⁺, or Fe²⁺, while Ni²⁺ was

less effective and Cu²⁺ inactive. For other metalloproteases, similar substitution effects have been observed[30,31]. Surprisingly though, the reaction speed was dramatically increased in the presence of Co²⁺ ($k_{cat} = 17.8$ min⁻¹), whereas the CsUb-binding efficiency appeared largely unaffected ($K_M = 17.8$ μM; Fig. 5b). Apparently, CsRpn11$^{\Delta 149-202}$ is rather flexible in its use of metal ions in vitro. When it comes to metalloprotein formation in vivo though, organisms are known to be highly selective in the choice of their enzyme ligands, employing elaborate mechanisms to ensure metal ion specificity[32,33]. Zn²⁺ is generally considered to be the natural ligand for other JAMM proteases[34]. Nonetheless, as *C. subterraneum* has been found in a subsurface hot water stream emanating from an underwater gold mine, enriched in a variety of transition metals[35], the physiological use of Co²⁺ cannot be excluded.

**Inhibition of CsRpn11 and CsJAMM by thiolutin and 8-TQ**. Recently, the metal-binding ability of Rpn11 has become a focus of attention due to the discovery of Rpn11/JAMM-specific inhibitors, which act by coordinating the catalytic Zn²⁺. A library screen of metal-binding compounds identified 8-thioquinoline (8-TQ) and derivatives as potent inhibitors. They were highly selective for Rpn11, as most other eukaryotic metalloproteases, including AMSH, were not affected[36]. On the other hand, the antibiotic thiolutin and structurally related dithiolopyrrolones were found to have a more relaxed specificity and inhibited eukaryotic JAMM proteases in general[37]. We were curious to see the effects of these inhibitors on prokaryotic JAMM proteases and added them to CsRpn11 and CsJAMM at concentrations which elicit significant responses in eukaryotes. Indeed, CsUb-pre processing by CsRpn11 was strongly affected by both 8-TQ and thiolutin (Fig. 5c), whereas effects on CsJAMM-mediated CsSAMP processing were significantly weaker (Supplementary Fig. 8b). These results underscore the close relationship between archaeal and eukaryotic Rpn11 (Fig. 1).

**Cleavage site effects on the processing of CsUb conjugates**. CsRpn11 can cleave linearly extended Ub proteins, such as Ub-pre (see above), artificial linear fusions such as CsUb-GFP and a non-physiological isopeptide-bond-conjugated protein, Rad50[11]. Also eukaryotic Rpn11 is able to process linear amide bonds[28]. The cleavage site in an isopeptide-linked conjugate potentially poses a steric challenge for processing, as it might be affected by the folding state of the conjugated substrates. The extended Ub C-terminal tail could be a means to mitigate steric problems and to permit efficient binding of DUBs, including CsRpn11. In this context, the intriguing conservation of the C-terminal Gly–Gly motif in Ub-like proteins might be of relevance, too. To address the issue of conformational constraints at the cleavage

**Fig. 3** Crystal structures of the deubiquitinase CsRpn11 and of the CsRpn11–CsUb complex from the archaeon *C. subterraneum*. **a** Structure-based sequence alignment of CsRpn11, *S. cerevisiae* Rpn11 (ScRpn11), and *H. sapiens* AMSH-LP (HsAMSH-LP). The color scheme is as in Fig. 2, except that catalytic residues are wrapped in black boxes and that ScRpn11 Gly77 and equivalent residues are highlighted in magenta. Ins-1 and Ins-2 regions (see main text) are shown in cyan, the Ins-1 β-sheets induced by CsUb binding are indicated by arrows. AMSH Ins-2 and C terminus differ from the corresponding elements in Rpn11 and are not shown in the alignment. **b** The crystal structures of CsRpn11 and CsRpn11–CsUb, with the active site zinc ion shown in black. The structure of C-terminally truncated eukaryotic Rpn11 from *S. cerevisiae* (PDB 4O8X) is shown for comparison. The Ins-2 region, which is not resolved in the *S. cerevisiae* Rpn11 structure, is modeled based on the structure of proteasome-associated human Rpn11 (PDB 5T0C). Residues 174–202, including H5, are not resolved in the CsRpn11 structure and the corresponding region is also absent from the truncated *S. cerevisiae* Rpn11 structure. CsUb is colored gray. **c** Close-up views of the active site in CsRpn11 crystal structures illustrate the catalytic mechanism[16]. The CsUb-free CsRpn11 structure (left; PDB 6FJU) represents the substrate-accepting state. The transition state is modeled: Upon CsUb binding, the catalytic water performs a nucleophilic attack on the Gly78 carbonyl carbon (middle left; CsRpn11–CsUb (PDB 6FNO), shown with the Ub backbone of AMSH-Ub₂ (PDB 4NQL[16]) aligned to CsUb (not shown)), yielding processed CsUb (middle right; PDB 6FNN). Binding of a new catalytic water molecule regenerates the active site and possibly facilitates CsUb dissociation (right; PDB 6FNO). **d, e** Critical interactions between CsRpn11 and CsUb in the same orientation as in **b**. Shown are hydrophobic contacts between CsUb and CsRpn11, and the backbone coordination of the CsUb C terminus that results in β-sheet formation in the Rpn11 Ins-1 site. For simplicity, only side chains that are involved in these contacts are shown

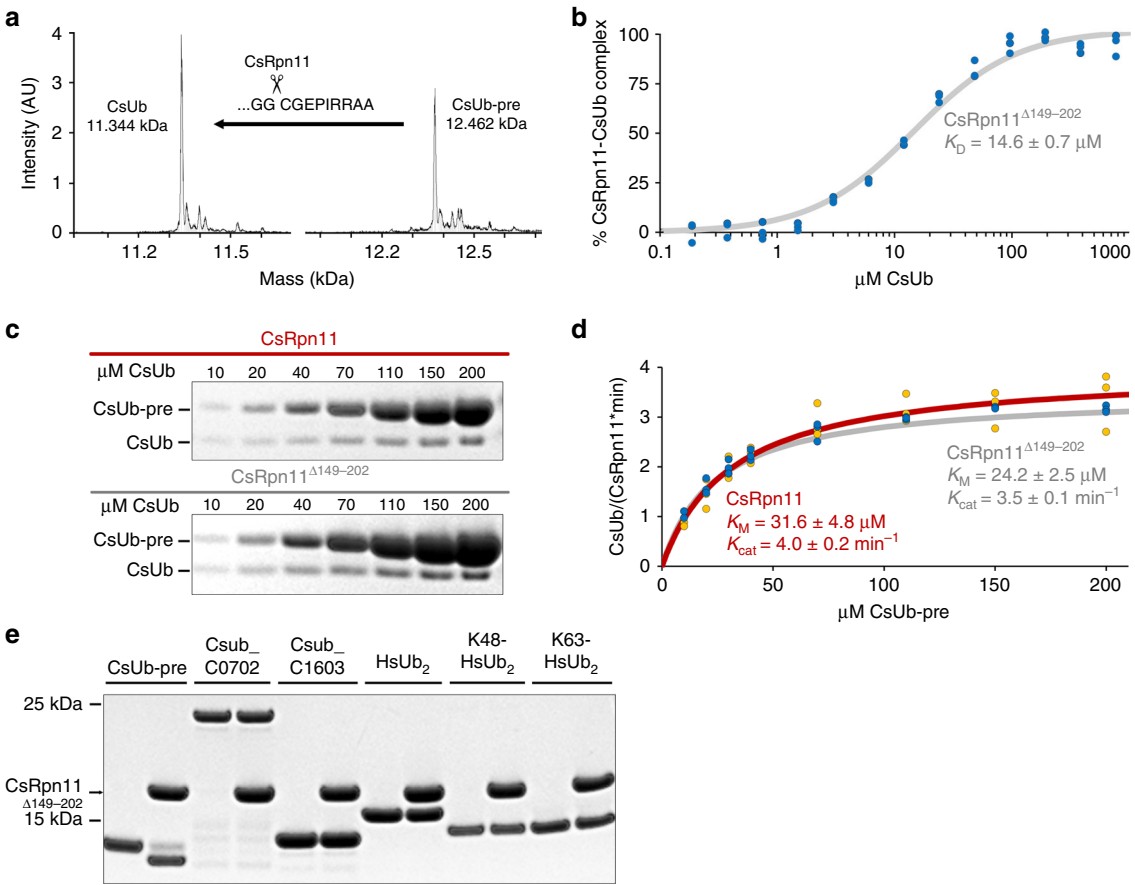

**Fig. 4** CsRpn11 specifically processes CsUb precursor (CsUb-pre) at the Gly–Gly motif. **a** Mass spectrometric profile of CsUb-pre before and after incubation with CsRpn11. The mass of the reaction product, mature CsUb (CsUb), corresponds to CsUb-pre without the nine amino acid fragment succeeding the Gly–Gly motif. All masses refer to N-terminally His$_6$-tagged CsUb (see methods). **b** Microscale thermophoresis (MST)-binding curve for $K_D$ determination of the CsRpn$^{\Delta149-202}$–CsUb interaction based on three independent experiments. **c, d** Sample gels used for $K_M$ determination of CsUb-pre processing by full-length and truncated CsRpn11. The resulting Michaelis–Menten plot is based on three independent experiments. **e** SDS-PAGE analysis of CsRpn11 substrate specificity. CsRpn11 cleaves CsUb-pre but is inactive toward *C. subterraneum* SAMP precursors Csub_C0702 and Csub_C1603 (CsMoaD), as well as isopeptide- (Lys48 or Lys63) and linearly linked human diubiquitin (HsUb$_2$). The gel shows two lanes for each of the indicated substrates, before (left) and after (right) treatment with CsRpn11$^{\Delta149-202}$. The complete gels are shown in Supplementary Fig. 7

site, we first produced a suitable CsRpn11 substrate via an intein fusion technique[38] (Fig. 6), as physiological *C. subterraneum* Ub conjugates are not available. In short, an activated CsUb–Cysteamine conjugate was chemically ligated to exposed cysteine residues in the model substrate TEV protease[39]. The resulting conjugate differs from native isopeptide linkages only by substitution of the lysine γ-methylene group with a disulfide bridge (Fig. 6) and has been shown to effectively mimic isopeptide bonds in ubiquitination reactions[40,41]. This procedure generated TEV proteases conjugated to single CsUb moieties at either one or two of its Cys residues (CsUb-TEV and CsUb$_2$-TEV). When we treated this mixture with CsRpn11 and visualized the time course of the reaction by SDS-PAGE (Fig. 6), unconjugated CsUb and TEV protease concentrations increased over time, indicating that CsRpn11 effectively cleaves these isopeptide-linked CsUb conjugates.

In the CsRpn11–CsUb co-structure, CsRpn11 specifically interacts with C-terminal CsUb residues up to Gly78, after which the protease cleaves (Fig. 3d and Supplementary Fig. 6). If a variety of isopeptide-linked CsUb conjugates can get cleaved by CsRpn11, then the only common residue distal of the cleavage site is the Lys forming the isopeptide bond. To find out if CsRpn11 is oblivious to the region distal of the cleavage site, we produced a series of mutations in CsUb-pre and tested their

processing. Surprisingly, a Gly78Ala mutant was still processed by CsRpn11, indicating that the highly conserved Gly–Gly motif is not strictly required for deubiquitination (Fig. 7a). However, larger residues in position 78, such as Val or Glu, prevented cleavage, potentially through interference with CsRpn11 Thr101. The CsRpn11–CsUb complex structure (Fig. 3d) suggests the penultimate CsUb residue Gly77 to be more restricted than Gly78, which could make it even less tolerant to substitutions with larger residues. Indeed, a Gly77Ala mutant was processed at much slower rate than Gly78Ala, and a Gly77Val mutation prevented the reaction completely (Fig. 7a). Intriguingly, the efficiency of processing can also be increased, as demonstrated for the Cys79Gly and Glu81Trp mutants (Fig. 7a), which are cleaved at a faster rate than wild-type CsUb-pre. On the other hand, steric accessibility of the cleavage site appears to be a critical factor, as replacement by bulky residues in Cys79Glu or Gly80Trp mutants prevented CsUb-pre processing altogether (Fig. 7a). We conclude that residues distal of the cleavage site are being selected for steric compatibility and are able to significantly influence Ub-pre processing efficiency.

**Cleavage of CsUb conjugates is affected by their folding state.** Based on the results above, it seemed conceivable that CsRpn11-mediated deubiquitination is also affected by the conformation of

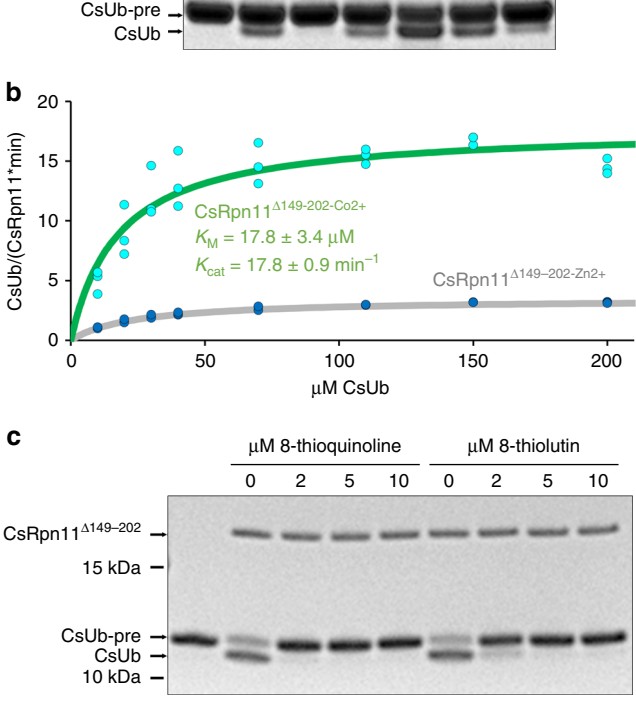

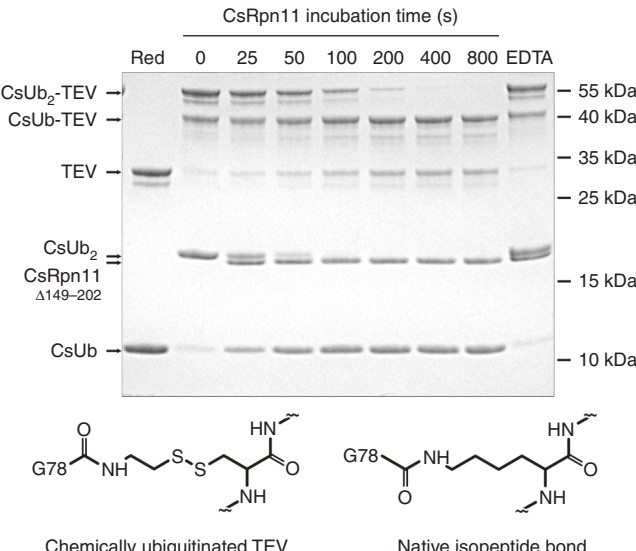

**Fig. 5** CsRpn11 activity is affected by the catalytic metal ion type and inhibitors thiolutin and 8-TQ. **a** CsUb-pre cleavage by CsRpn11$^{\Delta149-202}$ in the presence of the indicated metals. The reaction was analyzed by SDS gel electrophoresis. **b** Michaelis–Menten plot of CsUb-pre cleavage by Co$^{2+}$-bound CsRpn11$^{\Delta149-202}$ in comparison with the Zn$^{2+}$-bound enzyme (gray, see Fig. 4d). The curves are based on three independent experiments. **c** An aliquot of 1.75 μM CsRpn11$^{\Delta149-202}$ was incubated with the indicated concentrations of the Rpn11-specific inhibitors 8-TQ[36] or thiolutin[37] and tested for activity toward CsUb-pre

**Fig. 6** CsRpn11 cleaves isopeptide-bonded CsUb conjugates. Time course of CsRpn11$^{\Delta149-202}$-mediated cleavage of chemically ubiquitinated TEV protease with an isopeptide-bond-like linkage. The disulfide bridge in the linker is reducible (lane 1), pre-incubation with the chelator EDTA inhibits the reaction (lane 9). CsUb$_2$ is a byproduct of the CsUb-TEV preparation, which is also processed by CsRpn11. A comparison of the cleavage site in chemically ubiquitinated TEV to classical isopeptide bonds is shown below the gel[41]

ubiquitinated substrates at the ubiquitination site. To investigate this possibility, we fused CsUb to α-Lactalbumin. This protein forms a proteinase K-sensitive molten globule under reducing conditions, but is stabilized by four disulfide bridges under oxidizing conditions, rendering the protein largely proteinase K-resistant (Fig. 7b)[42]. CsRpn11 deubiquitinated both α-Lactalbumin forms with comparable efficiency (Fig. 7b), indicating that the CsUb cleavage site was fully accessible to CsRpn11, irrespective of the α-Lactalbumin folding state. In a related experiment, we produced a CsUb fusion with mouse dihydrofolate reductase (DHFR). Apo-DHFR is a rather thermolabile enzyme that can be stabilized by its ligands NADPH and methotrexate[43,44]. Thus, at the assay temperature of 45 °C, a significant fraction of unliganded DHFR is (temporarily) unfolded, while the Holo-form is still unaffected (Fig. 7c). CsRpn11 could only process the Apo-form under these conditions, indicating that partial protein unfolding is required for deubiquitination of this substrate (Fig. 7c). However, when we introduced a short-linker region between CsUb and DHFR, CsRpn11 did no longer discriminate between folded and unfolded DHFR, suggesting that spatial constraints were indeed responsible for this distinction (Fig. 7c).

**Archaeal ubiquitination in an evolutionary context**. While Ub conjugation was long thought to be restricted to eukaryotes and to have evolved from distant prokaryotic homologs, such as ThiS or MoaD[3], the bioinformatic discovery of complete Ub modification systems in *C. subterraneum*, and more recently in Asgard archaea, suggests that the fundamental function of this system, protein tagging, was already established in archaea. The first true archaeal Ub modification systems, however, co-existed side by side with the universal archaeal SAMP/JAMM system, making it necessary to avoid cross-reactivity. For the CsRpn11/CsUb and CsSAMP/CsJAMM pairs this probably entailed the exclusive recognition of their respective partners, as we observe it for CsRpn11 and CsJAMM (Fig. 4e and Supplementary Fig. 8a). We have shown that CsRpn11 is highly active on its own, cleaving both CsUb precursors and, at a later stage of the ubiquitination cascade, CsUb-conjugated substrate proteins (Fig. 6). Cleavage of Ub precursors in extant eukaryotes is carried out by DUBs, a large class of proteins that evolved for deubiquitination processes in the cytosol[29] and which appears to be absent in archaea. The eukaryotic Rpn11 probably ceased to cleave ubiquitinated proteins in solution when it became incorporated into the proteasome and instead became dependent on Ub-conjugated substrate delivery at the 19S lid. It has been proposed that the ensuing deubiquitination is coupled to substrate unfolding and protein degradation[45–47]; mechanical substrate pulling into the proteasomal AAA ATPase motor is thought to facilitate the transition of the Rpn11 Ins-1 motif from the inactive closed state to the hairpin of the active enzyme form[18]. For CsRpn11, we see an equivalent conformational transition in Ins-1 induced by CsUb-binding alone (Supplementary Fig. 5), in the absence of auxiliary factors, indicating that this is an ancestral feature, rather than a mechanistic adaptation of eukaryotic Rpn11 to its incorporation into the proteasomal lid.

As the demands for a regulated protein quality-control system increased in parallel with the evolution of eukaryotes, the Ub system gained in complexity too. Considering the staggering number of proteins involved in the eukaryotic ubiquitination machinery, one can ask how an archaeal system could possibly operate with just a few components, with for example only two small E3 RING finger proteins in *C. subterraneum*, compared to hundreds of sophisticated E3 enzymes in eukaryotes[48]. However, proteins in related archaeal systems are known to be remarkably

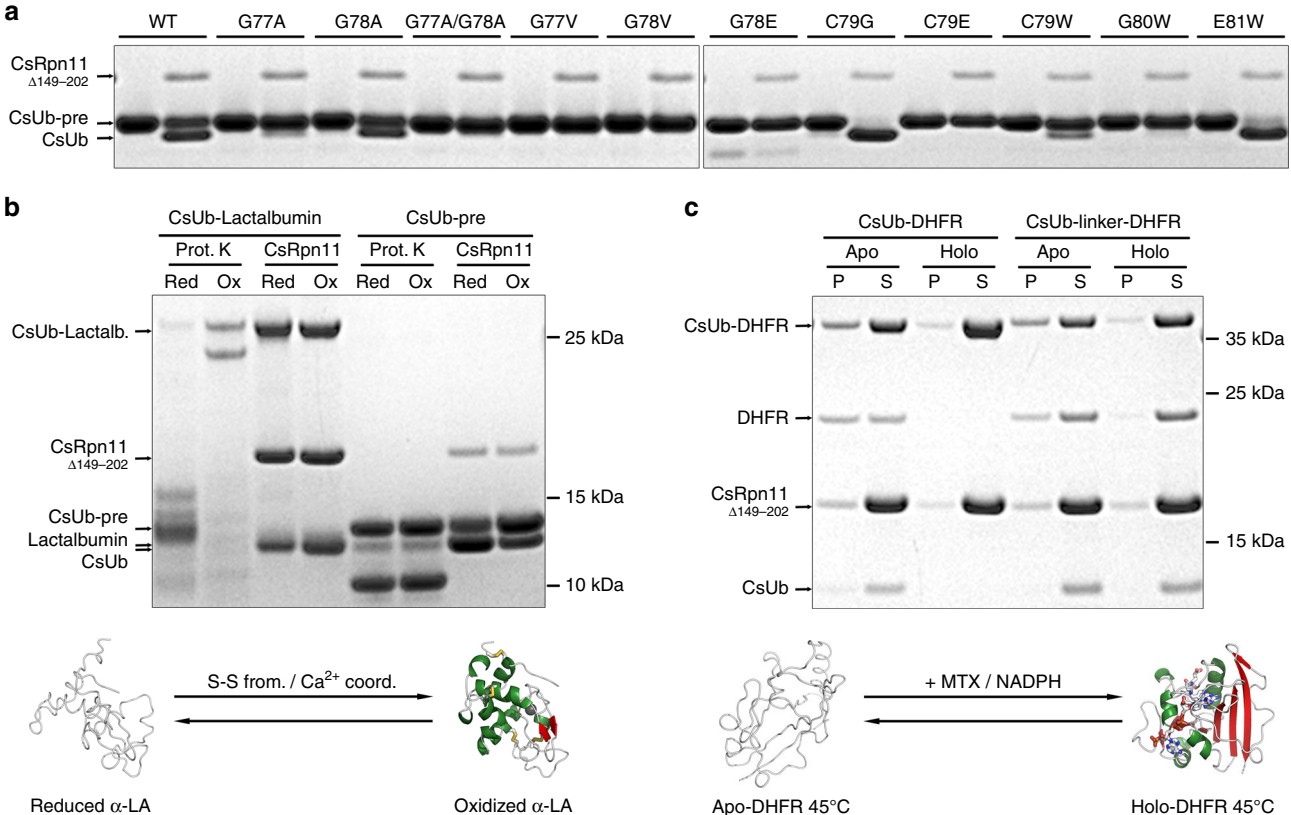

**Fig. 7** Effects of CsUb-pre cleavage site mutations and the conformation of conjugated substrate on processing by CsRpn11. **a** Protein gels showing CsUb-pre mutants, each before and after CsRpn11$^{\Delta149-202}$ treatment. **b** Processing of a CsUb–Lactalbumin (α-LA) fusion by CsRpn11$^{\Delta149-202}$ and proteinase K under reducing and oxidizing conditions. These conditions have only minor effects on the general activity of both proteases, as shown by the control digest with CsUb-pre alone. **c** Processing of a CsUb–DHFR fusion by CsRpn11$^{\Delta149-202}$ in the presence (Holo) or absence (Apo) of the stabilizing DHFR interactors MTX and NADPH[43]. At the assay temperature, Apo-DHFR unfolds and is found partially in the pellet (P), while Holo-DHFR is detected only in the supernatant (S)

versatile. For example, *H. volcanii* E1 activates SAMPs for modification of protein functions, for sulfur transfer and for use as a specialized degradation tag[49]. In a similar fashion, we have shown that a protein like CsRpn11 can pursue multiple tasks, which in modern eukaryotes are distributed among several, more specialized, proteins. While our structural and biochemical results have provided insights into CsRpn11-mediated processing of CsUb-pre and deubiquitination of CsUb protein conjugates, the processes leading to CsUb conjugation are unknown. It will be of central importance to learn how substrates are selected for ubiquitination, and how tagging is possibly coupled to substrate degradation.

## Methods

**Bioinformatics**. To gather sequences of Rpn11 homologs, we searched the non-redundant protein sequence database at NCBI, comprising either bacterial, archaeal, or eukaryotic proteins, employing at least four iterations of PSI-BLAST[50] at default settings. JAMM core domains of the following proteins were used as seeds for the first iteration of these searches: *Homo sapiens* Rpn8 and Rpn11, *Haloferax volcanii* JAMM-1 and JAMM-2, and *Frankia alni* JAMM. The sequences in these sets were then clustered by their all-against-all pairwise BLAST *P* values in CLANS[19] to identify individual JAMM protein types and their relationships. Based on this preliminary map, the ten most distinct sequences of each cluster were extracted and used in further rounds of iterative PSI-BLAST searches to identify all members of the respective protein type. After each iteration, sequences to be included for the next iteration were manually reviewed. Next, we pooled together all sequences, aligned them based on homologs with three-dimensional structures using PROMALS3D[51], and truncated all elements N- and C-terminal of the JAMM core domain. To further decrease redundancy, we filtered the sequences down to a maximum pairwise identity of

70% and clustered them in CLANS to generate the cluster map shown in Fig. 1a. Clustering was done to equilibrium in 2D at a BLAST *P* value cutoff of 1e−15 and the final cluster map was made by showing all connections with a *P* value better than 1e−10.

The Ub cluster map (Fig. 1b) was generated using a similar approach, except that the initial PSI-BLAST search was seeded with *H. sapiens* Ub, SUMO, UFM1, and Urm1, and ThiS and MoaD from *E. coli* and *H. volcanii*. Clustering was done at a BLAST *P* value cutoff of 9e−6 and the final cluster map was made by showing all connections with a *P* value better than 1e−4.

The structure-based sequence alignments in Figs. 2 and 3 were generated with PROMALS3D, based on the structures of close homologs, and then manually refined. Structures were visualized using PyMOL v1.8.0.5, and DaliLite v3[52] was used for calculating structural alignments and similarity scores.

**Protein expression and purification**. *C. subterraneum* genes *Csub_C0702* and *Csub_C1603* (SAMPs), *Csub_C1473* (CsRpn11), *Csub_C0703* (CsJAMM), and *Csub_C1474* (CsUb) were synthesized (Eurofins) for expression in *E. coli* (Supplementary Table 3). Genes were cloned into a modified pET30b expression vector using *Nde*I and *Hin*dIII restriction sites, so that resulting proteins were produced with N-terminal His$_6$ tags and TEV protease cleavage sites (Supplementary Table 4).

The CsRpn11 vector was the basis for PCR-based site-directed mutagenesis to obtain a series of CsRpn11 mutants with deletions in the sequence C-terminal of the JAMM domain, CsRpn$^{\Delta175-202}$, CsRpn$^{\Delta167-202}$, CsRpn$^{\Delta149-202}$, and CsRpn$^{\Delta145-202}$. In a similar fashion, the CsUb vector was used to generate vectors for expression of CsUb-pre mutant genes G77A, G78A, G77A/G78A, G77V, G78V, G78E, C79G, C79E, C79W, G80W, and E81W. CsUb$^{C33A}$ was generated by round-the-horn mutagenesis[53]. Via fusion PCR, CsUb was ligated to either *Mus musculus* DHFR, mature *Bos taurus* Lactalbumin, or GyrA intein to produce His$_6$-CsUb–Lactalbumin, His$_6$-CsUb–DHFR, and CsUb$^{C33A}$–GyrA fusions. A variant of the CsUb–DHFR fusion contained a three amino acid linker, Cys–Gly–Glu, between both domains.

*E. coli* BL21 gold cells (Thermo) were transformed with the respective plasmids and grown at 25 °C in a lysogeny broth (LB). Protein expression was induced with 0.5 mM isopropyl-β-D-thiogalactoside at an optical density of 0.4 at 600 nm. Cells were harvested after 16 h, lysed by French Press, and subjected to ultracentrifugation. To purify insoluble proteins (all CsRpn11 variants, Lactalbumin fusions), the ultracentrifugation pellet was washed with buffer A (50 mM HEPES-NaOH pH 7.5, 250 mM NaCl, 10 mM DTT, 1 M urea, 1% Triton X-100), resuspended in buffer B (50 mM Tris-HCl pH 8.0, 250 mM NaCl, 10 mM imidazol, 8 M urea) and purified using HisTrap HP columns (GE Healthcare). CsRpn11 variants were then refolded via dialysis against buffer C (20 mM HEPES-NaOH pH 7.5, 500 mM NaCl) at room temperature, and Lactalbumin fusions by dialysis against buffer D (20 mM HEPES-NaOH pH 7.5, 150 mM NaCl, 2.5 mM glutathione (GSH), 0.5 mM glutathione disulfide (GSSG), 1 mM CaCl$_2$). Soluble proteins were purified via HisTrap HP columns (20 mM Tris-HCl pH 8.0, 250 mM NaCl, 0–250 mM imidazole). CsUb–DHFR and CsUb$^{C33A}$–GyrA were subsequently purified via Superdex 75 10/300 GL columns (GE Healthcare) using buffer C. Wild-type and mutant CsUb proteins were not purified via gel filtration and instead dialyzed against buffer C after a 10 min heat incubation at 70 °C.

**Crystallization and data collection**. Crystallization trials were performed in 96-well sitting-drop plates, with drops typically containing 300 nl of reservoir solution and 300 nl of protein solution, and a reservoir of 50 μl. The composition of the protein solutions, the successful reservoir solutions, and the cryo protectant solutions are summarized in Supplementary Table 5. For the zinc soak structure, prior to cryo protection, CsRpn11$^{Δ149-202}$–CsUb crystals were soaked for 24 h in reservoir solution supplemented with 5 mM ZnCl$_2$. Data were collected at 100 K using a PILATUS 6M-F hybrid pixel detector (Dectris) at beamline X10SA at the Swiss Light Source (Villigen, Switzerland). The wavelength was set to 1.283 Å for the CsRpn11$^{Δ149-202}$–CsUb structure, otherwise to 1 Å. All data were indexed, integrated, and scaled using XDS[54].

**Crystal structure determination**. All structures were solved via molecular replacement using MOLREP[55]. The search model for the CsUb structure was 1UBQ, and 4OCL for the full-length CsRpn11 structure. The refined coordinates of these structures were subsequently used as search models to solve the structures of CsRpn11$^{Δ149-202}$ and CsRpn11$^{Δ149-202}$–CsUb. The structure of the CsRpn11$^{Δ149-202}$–CsUb zinc soak was solved on the basis of the un-soaked CsRpn11$^{Δ149-202}$–CsUb structure. After molecular replacement, the structures of CsUb and full-length CsRpn11 were rebuilt using Buccaneer[56]. All structures were completed by cyclic manual modeling with Coot[57] and refinement with REFMAC5[58]. Data collection and refinement statistics are given together with PDB accession codes in Supplementary Table 6.

**Biophysical methods**. Static light-scattering experiments (SEC-MALS) were performed with 2 mg ml$^{-1}$ His$_6$-tagged CsRpn11 in buffer C, using a Superdex S200 10/300 GL gel size-exclusion column (GE Healthcare) coupled to a miniDAWN Tristar Laser photometer (Wyatt) and a RI-2031 differential refractometer (JASCO). Data analysis was carried out with ASTRA V software (Wyatt).

For liquid chromatography mass spectrometry (LCMS) measurements, 10 μM CsUb-pre were incubated with 1 μM CsRpn11 in buffer C for 20 min at 45 °C and compared to an untreated CsUb-pre sample. Desalted samples were subjected to a Phenomenex Aeris Widepore 3.6 μm C4 200 Å (100 × 2.1 mm) column, eluted with a 30–80% H$_2$O/acetonitrile gradient over 15 min in the presence of 0.05% trifluoroacetic acid, and analyzed with a Bruker Daltonik microTOF. Data processing was performed in Compass DataAnalysis 4.2 and the m/z deconvoluted to obtain the protein mass via MaxEntropie.

For microscale thermophoresis (MST) experiments, measuring protein–protein interactions and affinities, purified CsRpn11$^{Δ149-202}$ was labeled with flurophore using the NT-647-NHS labeling kit (Nanotemper). Next, a serial 1:1 dilution of CsUb ranging from 47 nM to 770 μM was prepared and mixed with 50 nM labeled CsRpn11$^{Δ149-202}$. MST measurements were performed at a Monolith NT.115 (Nanotemper), using various MST power and laser intensity settings to test the general validity of the obtained data. The results shown in Fig. 4b represent mean values obtained in three independent experiments and were measured at a temperature of 45 °C, using MST power 80% and laser intensity 40%. The shown binding curve was fitted to the data, using the NT Analysis 1.5.41 software (Nanotemper).

Thermal denaturation curves to monitor protein stability were recorded by circular dichroism spectroscopy at 220 nm using a JASCO J-810 spectropolarimeter.

**Biochemical methods**. To analyze the effect of the active site metal ion on catalysis, CsRpn11$^{Δ149-202}$ was incubated with an excess of EDTA (2 mM) and dialyzed against buffer C, using trace metal analysis grade reagents. An aliquot of 1.75 μM of the resulting catalytically inactive Apo-CsRpn11$^{Δ149-202}$ sample was then supplemented with 40 μM of either CuCl$_2$, NiCl$_2$, CoCl$_2$, FeCl$_2$, or MnCl$_2$, and

incubated with 14 μM CsUb-pre at 45 °C for 5 min. The reaction was stopped with 1 mM EDTA, and cleavage efficiency of the samples was analyzed by SDS-PAGE.

To investigate the substrate specificities of CsRpn11 (Fig. 4e) and CsJAMM (Supplementary Fig. 8a), 0.2 mg ml$^{-1}$ CsUb-pre, Csub_C0702, Csub_C1603, HsUb$_2$ (Enzo Scientific, BML-UW0775), K48-linked HsUb$_2$ (Enzo Scientific, BML-UW9800), or K63-linked HsUb$_2$ (Enzo Scientific, BML-UW0730) were incubated with 0.2 mg ml$^{-1}$ CsRpn11$^{Δ149-202}$ or CsJAMM as indicated. Reactions were performed for 15 min at 25 °C, stopped with 1 mM EDTA, and analyzed by SDS-PAGE.

For the CsRpn11/CsJAMM inhibition assays (Fig. 5c and Supplementary Fig. 8b), 1.75 μM CsRpn11$^{Δ149-202}$ or CsJAMM were incubated at room temperature for 10 min with the indicated concentrations of either thiolutin, 8-thioquinoline (Sigma), or their respective solvents (DMSO or EtOH) in buffer C, before they were allowed to react with 14 μM CsUb-pre or SAMP Csub_C0702 at 45 °C for 8 min. The reaction was stopped with 1 mM EDTA, and cleavage efficiency of the samples was analyzed by SDS-PAGE.

For kinetic measurements at 45 °C (Figs. 4d and 5b), the indicated CsUb-pre concentrations were incubated for 21 min in buffer C with 400 nM Zn$^{2+}$-bound CsRpn11, 400 nM Zn$^{2+}$-bound CsRpn11$^{Δ149-202}$, or 76 nM Co$^{2+}$-bound CsRpn11$^{Δ149-202}$. Kinetic measurements at 60 °C (Supplementary Fig. 9) were performed for 5 min with 1 μM enzymes. Cleavage was stopped with 1 mM EDTA and samples loaded on 12% Nu-PAGE Bis-Tris gels (Thermo). Gels were stained with Coomassie blue G-250. The resulting band intensity was quantified using ImageJ and compared to CsUb standards on the same gel (0.5, 1.0, and 1.5 μg). The results shown in Figs. 4d and 5b represent mean values of three independent experiments. The binding curve was fitted to the data using the SigmaPlot 12.3 enzyme kinetics tool with the Michaelis–Menten model.

The effect of amino acid substitutions in the CsUb C terminus was assayed by incubating 14 μM of the respective CsUb-pre mutants with 1.75 μM Zn$^{2+}$-bound CsRpn11$^{Δ149-202}$ in buffer C at 45 °C for 8 min. Reactions were stopped with 1 mM EDTA and cleavage efficiency of the samples compared by SDS-PAGE.

The effect of the substrate folding state was assayed with CsUb–Lactalbumin and CsUb–DHFR fusions. An aliquot of 6.15 μM CsUb–DHFR fusions was processed with 14 μM CsRpn11$^{Δ149-202}$ for 5 min at 45 °C in the presence or absence of 440 μM methotrexate and NADPH ligands. The reaction was stopped with 1 mM EDTA and soluble and insoluble fractions separated via centrifugation at 20,000 × *g* for 3 min. CsUb–Apo-Lactalbumin was obtained by reducing the refolded Holo-form (see protein purification) with 10 mM TCEP and 10 mM EGTA, followed by dialysis against buffer E (20 mM HEPES, 150 mM NaCl, 5 mM TCEP, 0.5% Chelex 100 resin [BioRad]). An aliquot of 12 μM of both CsUb–Lactalbumin forms or 14 μM of CsUb-pre was then incubated with either proteinase K or CsRpn11$^{Δ149-202}$ under oxidizing (buffer D) or reducing (buffer E) conditions. Proteinase K digests were performed with 0.1 μg ml$^{-1}$ enzyme for 30 min at 10 °C and stopped with 4 mM PMSF. CsRpn11$^{Δ149-202}$ digests were performed at 45 °C for 17 min with 14 μM enzyme in case of CsUb–Lactalbumin and for 8 min with 1.75 μM enzyme in case of CsUb.

To obtain an isopeptide-linked ubiquitin conjugate[38], a CsUb$^{C33A}$–GyrA intein fusion was spliced by incubation with 50 mM cystamine-dihydrochloride, 50 mM Tris(2-carboxyethyl) phosphine for 48 h at 23 °C in buffer F (50 mM Tris pH 7.8, 100 mM NaCl, 1 mM EDTA). The activated CsUb$^{C33A}$ was chemically modified with a 20× molar excess of dithionitrobenzoate (30 min at 23 °C in buffer F) and afterward added in 4× molar excess to purified TEV protease as substrate. The conjugation reaction took place for 1 h at 23 °C in buffer G (20 mM Tris pH 7, 50 mM NaCl, 1 mM EDTA), followed by purification of the conjugate by gel-sizing chromatography (Superdex 75, buffer C). To probe CsRpn11-mediated cleavage, 0.25 mg ml$^{-1}$ of the reaction product was incubated with 3.33 μM CsRpn11$^{Δ149-202}$ in buffer C at 37 °C and the reaction stopped with 1 mM EDTA at the indicated times. The results were compared to a reaction in presence of 1 mM EDTA for the maximum incubation time (800 s) via SDS-PAGE.

**Data availability**. Protein structure coordinates and structure factors have been deposited in the Protein Data Bank under accession codes PDB 6FJ7 (CsUb), PDB 6FJU (CsRpn11), PDB 6FJV (CsRpn11$^{Δ149-202}$), PDB 6FNN (CsRpn11$^{Δ149-202}$–CsUb), and PDB 6FNO (CsRpn11$^{Δ149-202}$–CsUb zinc soak). All relevant data are available from the corresponding author upon request.

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

## Acknowledgements

We thank Andrei Lupas for discussions and continuous support, Reinhard Albrecht for setting up crystallization screens and diffraction experiments, Marc Claassen (Max Planck Institute for Developmental Biology) and the core facility at the Max Planck Institute for Biochemistry (Martinsried) for mass spectrometric analysis, and the staff of beamline X10SA/Swiss Light Source for excellent technical support. This work was supported by institutional funds from the Max Planck Society.

## Author contributions

A.C.D.F. and J.M. designed the project and the biochemical experiments; A.C.D.F., L.M. and M.W. performed the biochemical experiments; A.C.D.F. did the bioinformatics; M.D.H. did the crystallography; A.C.D.F., M.W., M.D.H. and J.M. analyzed the data;

A.C.D.F., M.D.H. and J.M. wrote the manuscript; and all authors reviewed the results and approved the final version of this manuscript.

## Additional information

**Competing interests:** The authors declare no competing interests.

