## [Peer Review File · Nature Communications]

Reviewers' comments:

Reviewer #1 (Remarks to the Author):

In this manuscript, Fuchs et al. report structural and mechanistic characterization of an archaeal JAMM-3 type metalloprotease Csub_C1473 (called "CsRpn11" in the manuscript), Csub_C1474 (called "CsUb-pre" in the manuscript), and Csub_C1473/Csub_C1474 complex, from *C. subterraneum*. New insights into biochemical, structural and substrate recognition features of Csub_C1473 are shown. One main point that this study claim is Csub_C1473 is archaeal Rpn11, however, I disagree. Reasons as follows:

First of all, the key feature/biological role of eukaryotic Rpn11 is its association with AAA ATPase and 20s core particle for ubiquitin-mediated protein degradation. This feature discriminates Rpn11 from other eukaryotic JAMM proteases, such as CSN5, AMSH, BRCC36. If without this feature, it is hard to think Csub_C1473 is Rpn11-like, rather than AMSH-like or CSN5-like or BRCC36-like?

Second, some of the evidence provided in this study doesn't support or even oppose Csub_C1473 is Rpn11-like. For example, Csub_C1473 is active independently and behaves as a monomer in solution, which makes Csub_C1473 resemble AMSH rather than Rpn11 (require association with protein partners for activity). For another example, Csub_C1473 ("CsRpn11") process Csub_C1474 ("CsUb-pre") into mature form, but authors also mention other eukaryotic DUBs rather than Rpn11 do this Ub-pre processing job. In addition, Csub_C1473 has substrate discrimination feature and does not cleave eukaryotic Ub2 (M1, K48 and K63 linkage), this appears indicating Csub_C1474/Csub_C1473 system separates from Ub/Rpn11 system.

To me, results in this study indicate Csub_C1473 is a novel JAMM-type protease in Archaea (for example, archaeal JAMM-3 type in Fig.1) and Csub_C1474 is another type of Ub-like protein in Archaea (for example, another type of SAMP). You can only draw conclusions based on what the results show, and further claim that Csub_C1473 is archaeal Rpn11 and Csub_C1474 is archaeal ubiquitin is not right. Treating "Csub_C1473/Csub_C1474" system as first true archaeal Ub modification system is inappropriate, especially when we consider archaeal SAMP system contains components with both sequence and function similarities to eukaryotic Ub, E1, E3, JAMM metalloprotease, AAA ATPase and 20S core particle (Humbard et al., 2010; Miranda et al., 2014; Miranda et al., 2011; Hepowit et al, 2016; Fu et al, 2017; Hepowit et al., 2012; Cao et al., 2017; Fu et al., 2016).

In addition, this manuscript is mainly about "the structure of an archaeal JAMM (Csub_C1473)/Ub-like protein (Csub_C1474) complex" as the title shows. However, this complex adds little to our understanding of JAMM proteases in general, as both "eukaryotic JAMM/Ub complex" and "archaeal JAMM/Ub-like protein complex" structures have been well characterized in the cases of eukaryotic AMSH/AMSH-LP (Sato., 2008; Shrestha et al., 2014), Rpn11 (Worden et al., 2017) and archaeal JAMM-1 (Cao et al., 2017).

1. Page6 "Remarkably, while prokaryotic JAMMs which cluster distant from Rpn11 are functionally associated with just a SAMP- or Ub-like protein and an E1-like homolog,"

Comment: Inaccurate. Not just a SAMP- or Ub-like protein, at least three Ub-like proteins reported (e.g., SAMP1, SAMP2 and SAMP3) and related with archaeal JAMMs.

2. Page6 "Finally, *C. subterraneum* and Asgard archaea, whose JAMM sequences are most similar to eukaryotic Rpn11, contain in addition RING-type E3-homologs to complete the ubiquitination machinery."

Comment: Don't agree. Need to modify this sentence.

First, "RING-type E3-homologs" is inappropriate, because CSUB_C1477 is only homolog of RING finger protein, a component of Cullin-RING E3 ligase.

Second, "to complete the ubiquitination machinery" is dangerous. For example, substrate receptor of E3 is missing. Although in vitro CsE1, CsE2 and CSUB_C1477 ubiquitinate non-physiological substrate (James et al., 2017), however, in vivo, other components (e.g. substrate receptor of E3) of ubiquitination machinery is probably needed to complete the ubiquitination of a substrate.

3. Page6 "is found distant from classical Ub and more closely related to prokaryotic SAMPs like Moad and ThiS."

Comment: Incorrect. No archaeal SAMPs like bacterial ThiS and archaeal SAMP1 is bi-functional. First, no archaeal SAMPs like bacterial ThiS of thiamine biosynthesis. Archaea lack steps of bacterial ThiF-ThiS-ThiG-ThiH pathway of synthesizing the thiazole ring, instead archaea apply eukaryote-like Thi4 mechanism to mobilize sulfur to form the thiazole ring (Hwang et al., 2017). Second, archaeal SAMP1 could be isopeptide-linked to hundreds of protein substrates, like eukaryotic ubiquitin, however, bacterial Moad doesn't have this feature. Archaeal SAMP1 is bi-functional, on one hand like eukaryotic ubiquitin for diverse protein substrates modification (Humbard et al., 2010; Dantuluri., 2016), on the other hand like bacterial Moad participating in MoCo biosynthesis (Miranda et al., 2011; Cao et al., 2015).

4. Page7 "CsUb is an extremely thermostable monomeric protein ($T_m > 95^\circ\text{C}$ in circular dichroism heat-denaturation studies, not shown)."

Comment: "not shown" is not ok. Other places in the manuscript also have this "data not shown" problems.

5. Page7 "In vivo, the protein is made as a precursor of 87 amino acids (Ub-pre), from which the last 9 residues are removed upon maturation. "

Comment: No data support. As you mentioned *C. subterraneum* could not be cultured so far, did you mean you co-express CsUb-pre (Csub_C1474) and CsRpn11 (Csub_C1473) in *E. coli* and find last 9 residues of CsUb-pre is removed by Csub_C1473 in *E. coli* as the in vivo assay? If so, show the data.

6. Page7 "CsUb also superimposes best with human Ub (DALI Z-Score 12.2), but is more distinct from so far characterized prokaryotic Ub-like proteins, such as *H. volcanii* Moad/SAMP1 ($Z = 6.9$) or ThiS/SAMP2 ($Z = 5.4$)."

Comment: Don't agree. *H. volcanii* SAMP2 is not ThiS-like and reasons were mentioned above. SAMP2 is eukaryotic ubiquitin-like, based on both functional and structural similarities, for example the following reasons:

First, SAMP2 could be isopeptide-linked to hundreds of protein substrates, like eukaryotic ubiquitin (Humbard et al., 2010).

Second, SAMP2 isopeptide modified protein substrate (e.g., TBP2) could be degraded in eukaryotic-like ubiquitin-proteasome pathway, in which archaeal SAMP2 (resemble eukaryotic ubiquitin), JAMM2 (resemble eukaryotic Rpn11), AAA ATPase PAN2, Cdc48c (resemble eukaryotic Rpt) and 20S proteasomes (resemble eukaryotic 20S proteasomes) were participated (Fu et al., 2016).

Third, SAMP2 could apply a structure conformation (when in complex with JAMM1) extremely similar to eukaryotic ubiquitin and is recognized by archaeal JAMM1 in a similar way as ubiquitin is recognized by eukaryotic JAMM (Cao et al., 2017).

How does the structure of Ub compare to CsUb vs. PfSAMP2 in complex with JAMM1? Did you compare the similarity of Ub vs. CsUb to Ub vs. PfSAMP2 in complex with JAMM1? If they don't show significant difference, then the statement "CsUb also superimposes best with human Ub but is more distinct from SAMP1 or SAMP2" is incorrect.

7. Page8 "The physiological relevance of these dimers is unclear, considering that for all determined JAMM structures the dimerization interfaces differ (Figure S2)."

Comment: Inappropriate. You only show 4 examples in the FigureS2 to claim "all determined JAMM structures" (much more than 4 examples). In addition, 3 of 4 examples that you show are not physiological-relevant dimers, since they behave as monomer in solution and just show artificial dimer in crystal.

8. Page8 "In structure superimpositions, CsRpn11 displays the highest similarity to *S. cerevisiae* Rpn11 (DALI Z-Score 21.1), but is less similar to so far determined structures of other prokaryotic JAMMs, such as *P. furiosus* JAMM-120 ($Z = 15.1$)."

Comment: How about "CsRpn11 (Csub_C1473)" similarities to eukaryotic CSN5, BRCC36 and AMSH/AMSH-LP? It is necessary to show. For example, if Csub_C1473 is also very similar to AMSH or even more similar to AMSH, is that indicate Csub_C1473 is AMSH-like rather than Rpn11-like. I do have some concerns, especially when consider: First, Csub_C1473 is active independently without requirement to form JAMM+/JAMM- heterodimer or larger complex, which is more like AMSH rather than Rpn11. Second, Asgard archaea do have AMSH-related ESCRT homologs (Zaremba-Niedzwiedzka et al., 2017).

9. Page 11 "Our results show that the Ub binding mode of AMSH, Rpn11 and CSN5 was already established in an archaeal ancestor"

Comment: Inappropriate. Ub-like SAMP2 binding mode of archaeal JAMM1 is extremely similar to Ub binding mode of eukaryotic JAMM and has been shown in previous study (Cao et al., 2017).

10. Page12 "In fact, homologs of eukaryotic DUBs have not been found yet in archaea."

Comment: Incorrect. Did you mean non-JAMM metalloprotease type DUBs (for example, cysteine protease type DUBs)? if so, write it clearly.

11. Page12 "Likewise, neither linearly linked human Ub (HsUb2), nor Lys48- or Lys63-isopeptide linked HsUb2 were suitable substrates (Figure 4E)."

Comment: This evidence appears opposing your statement that Csub_C1474 is CsUb-pre and Csub_C1473 is CsRpn11. It supports Csub_C1474 is another type/group of archaeal Ub-like proteins and Csub_C1473 is Csub_C1474-specific protease.

In contrast, archaeal JAMM1 is more like Rpn11 in its ability to cleave true Ub-Ub dimers (Lys48 or Lys63 linkage) (Cao et al., 2017), compared to the archaeal Csub_C1473 ("CsRpn11").

12. Page 12-13 "At 45°C, truncated CsRpn11 processed CsUb-pre with a KM of 24.2 μM and a kcat of 3.5/min and bound the reaction product with a KD of 14.6 μM. With a KM of 31.6 μM and a kcat of 4.0/min, the full-length version displayed similar reaction parameters, indicating that the CsRpn11 C- terminal helices are indeed not critically required for catalysis. "

Comment: Why perform at 45 °C? Is it optimized temperature for CsRpn11 activity or close to physiological temperature that *C. subterraneum* grow/live? If neither of them, it is hard to know whether "C- terminal helices" impact catalysis or not. For example, if 45°C is far away from optimized temperature of CsRpn11 activity, the difference of kinetics behavior of CsRpn11 (full length) and CsRpn11 (without C- terminal helices) may not be significant, but if tested at optimized temperature, the much higher difference may occur.

13. Page14 "Though to our knowledge metal ion specificity has not been tested for other JAMM proteases,"

Comment: Incorrect. Metal ion specificity was tested in HvJAMM1 before (Hepowit et al., 2012).

14. Page14 paragraph about "8-TQ inhibit CsRpn11 (Csub_C1473)"; Page15 "The susceptibility of CsRpn11, in particular for the highly selective 8-QT, thus underscores the close relationship between archaeal and eukaryotic Rpn11."

Comment: Further experiments needed. Please aware other eukaryotic JAMMs are also sensitive to 8-QT/capzimin (8-QT derivative) (Li et al., 2017), only based on Csub_C1473 susceptibility to 8-QT without comparison, it could not indicate Csub_C1473 close relationship to Rpn11. Besides, archaeal JAMM-1 may also be sensitive to 8-QT. Please consider the following questions and possibilities.

Whether Rpn11-specific inhibitor 8-QT also inhibits archaeal JAMM-1 type protease (e.g., PfJAMM1 or HvJAMM1)? If 8-QT has stronger inhibition effect to archaeal JAMM-1 compared to "Csub_C1473 (CsRpn11)", is that indicating archaeal JAMM-1 rather than "Csub_C1473 (CsRpn11)" is close to Rpn11?

How about Csub_C1473 sensitivity to eukaryotic CSN5-specific inhibitors (Schlierf et al., 2016; Altmann et al., 2017)? If "CsRpn11 (Csub_C1473)" is more sensitive to CSN5-specific inhibitor

than Rpn11-specific inhibitor, is that indicating Csub_C1473 is CSN5-like rather than Rpn11-like?

15. Page15 "This procedure generated mono- and di-ubiquitinated TEV proteases (CsUb-TEV and CsUb2-TEV)."

Comment: Does di-ubiquitinated TEV mean two CsUb as a chain linked to TEV or two mono-CsUb linked to TEV?

16. Page17 "Conclusions" paragraph

Comment: Too many "discussion" exist in this paragraph.

17. Page 17 "While Ub conjugation was long thought to be restricted to eukaryotes and to have evolved from distant prokaryotic homologs, such as the ThiS or Moad conjugation system."

Comment: Incorrect. ThiS or Moad systems are not conjugation systems, but SAMP and ubiquitin systems are. ThiS and Moad system are systems of thiamine biosynthesis and MoCo biosynthesis, respectively.

18. Page18 "bioinformatic discovery of complete Ub modification systems in *C. subterraneum*, and more recently in Asgard archaea, suggests that the fundamental function of this system, protein tagging, was already established in archaea. The first true archaeal Ub modification systems, however, co-existed side by side with the universal archaeal SAMP/JAMM system,"

Comment: This statement indicates "archaeal CsUb-pre (Csub_C1474)/CsRpn11 (Csub_C1473)" and "archaeal SAMP/JAMM system" are two different systems, which I don't agree. To me, the results of this study indicate Csub_C1474 is another type of ubiquitin-like protein in archaea (for example SAMP4), besides SAMP1, SAMP2 and SAMP3. Csub_C1473 is another type (JAMM-3 type) of JAMM proteases in archaea, besides JAMM1 and JAMM2.

This statement indicates "CsUb-pre (Csub_C1474)/CsRpn11 (Csub_C1473)" system is first true archaeal Ub modification systems, which I don't agree. Asgard archaea and *C. subterraneum* have eukaryotic E3-like component (RING-finger protein) is a main point that the authors think "CsUb-pre (Csub_C1474)" system is first true archaeal Ub modification systems. However, SAMP system also has eukaryotic E3-like component, for example, archaeal MsrA (Fu et al., 2017). Archaeal MsrA is the homolog of eukaryotic CRBN of E3 ubiquitin ligase CUL4-RBX1-DDB1-CRBN (Fischer et al., 2014), and archaeal MsrA participates in Ub-like SAMP conjugation to substrates (Fu et al., 2017), function also similar to eukaryotic E3.

19. Page19 "one can ask how an archaeal system could possibly operate with just a few components, with for example only two small E3 RING finger proteins in *C. subterraneum*, compared to hundreds of sophisticated E3 enzymes in eukaryotes"

Comment: Misleading, need to modify this statement.

First, RING finger proteins in eukaryotes are also limited (e.g., Rbx1, Rbx2), instead of hundreds. Substrate receptors of E3 in eukaryotes are "hundreds" or even more. A lot of substrate receptors of E3 in archaea including *C. subterraneum* may also exist.

Second, the claim "one can ask how an archaeal system could possibly operate with just a few components" is based on limited in vitro evidence, since the organism *C. subterraneum* can not be cultivated so far.

20. Figure 1 and Figure 1 legends

Comment: Need some modification based on reasons mentioned in previous comments. For one example, "Archaeal Moad" and "Archaeal ThiS" are inappropriate and misleading.

21. Figure 4

Comment: The Mass results appear wrong. Figure 4A show CsUb-pre 12.462kDa and CsUb 11.344 kDa. However, based on my calculation Csub_C1474 (CsUb-pre) of 87 amino acid MW: 9.11 kDa; CsUb of 78 amino acid MW: 8.15 kDa. If you have some specially treatment to "Csub_C1474 (CsUb-pre)", please write clearly.

In addition, no standard (e.g., CsUb or CsUb-pre with known amount) was loaded on the same gel, how could you quantify gel bands of CsUb or CsUb-pre for kinetics calculation?

22. Figure 6

Comment: Control needed (e.g., "EDTA-treated Rpn11 with substrate") to make sure the "CsUb" band is due to "Csub_C1473 (CsRpn11)" cleavage of substrate. As I concern: during incubation, a little portion of substrate may be reduced and CsUb band occur.

Reviewer #2 (Remarks to the Author):

The authors provide a biochemical and structural analysis of an archeal JAMM-type deubiquitinating enzyme (csRpn11), which is remarkably similar to its eukaryotic Rpn11 counterpart. The different csRPN1 structures described in this manuscript - with and without bound archeal ubiquitin - are representatives of the different catalytic steps of the DUB reaction. This manuscript is very interesting for the evolution of the ubiquitin pathway.

The work described is solid and well-executed; the representation and discussion of the results are clear and leave little to be desired. However, there are a few minor points that could be improved on.

- page 7, lower part: it is not correct to claim that eukaryotic ubiquitin has 'equivalent Lys residues' compared to csUb. As the authors correctly state later, the lysines are found at different positions, which is hardly reconcilable with their 'equivalence'.

-page 8, bottom line: what is meant by the term 'circular JAMM protease core' ? I can't see anything circular here.

- page 9ff, in connection to figure 3: The authors describe several structural features (helices, insert regions), which are not indicated in the structure. In general, I think that structure 3B should be rendered bigger and should contain labels for the insert regions and other features mentioned in the text.

- page 11, top, explains the contribution of particular residues by referring to the yeast ScRpn11 sequence. However, the alignment figure 3A shows only archeal and human Rpn11 sequences, making it very hard to appreciate discussions of "Gly-66 in ScRpn11". The authors should either use the residue numbering of human Rpn11 or alternatively include the yeast sequence in figure 3A.

- page 13, the authors write "...the linear substrate Ub-TAMRA.." I don't see why Ub-TAMRA should be considered a "linear substrate". The manuscript does not describe the source of the the Ub-TAMRA used by the authors, but the commercially available Ub-TAMRA has the Ub-cterminus bound to the epsilon-amino group of a lysine residue (which in turn is coupled to TAMRA). Thus, Ub-TAMRA emulates an isopeptide linkage rather than the peptide linkage found in linear chains. The authors should clarify this.

- Figure 4e: I wonder why in this gel, the linear HsUb2 runs at a much higher MW than the K48- and K63-linked diUb species. In our hands, all three human diUb species run (almost) at the same height. Since there is no MW ladder provided, it is not clear how big the apparent MW difference really is. Also, I did not find a source for the used diUb species in the materials section. This should be rectified.

We would like to thank both reviewers for their comments and constructive criticism. All points that were raised are addressed in the revised manuscript and explained in detail below.

Reviewer #1 (Remarks to the Author):

In this manuscript, Fuchs et al. report structural and mechanistic characterization of an archaeal JAMM-3 type metalloprotease Csub_C1473 (called "CsRpn11" in the manuscript), Csub_C1474 (called "CsUb-pre" in the manuscript), and Csub_C1473/Csub_C1474 complex, from C. subterraneum. New insights into biochemical, structural and substrate recognition features of Csub_C1473 are shown. One main point that this study claim is Csub_C1473 is archaeal Rpn11, however, I disagree. Reasons as follows: First of all, the key feature/biological role of eukaryotic Rpn11 is its association with AAA ATPase and 20s core particle for ubiquitin-mediated protein degradation. This feature discriminates Rpn11 from other eukaryotic JAMM proteases, such as CSN5, AMSH, BRCC36. If without this feature, it is hard to think Csub_C1473 is Rpn11-like, rather than AMSH-like or CSN5-like or BRCC36-like? Second, some of the evidence provided in this study doesn't support or even oppose Csub_C1473 is Rpn11-like. For example, Csub_C1473 is active independently and behaves as a monomer in solution, which makes Csub_C1473 resemble AMSH rather than Rpn11 (require association with protein partners for activity). For another example, Csub_C1473 ("CsRpn11") process Csub_C1474 ("CsUb-pre") into mature form, but authors also mention other eukaryotic DUBs rather than Rpn11 do this Ub-pre processing job. In addition, Csub_C1473 has substrate discrimination feature and does not cleave eukaryotic Ub2 (M1, K48 and K63 linkage), this appears indicating Csub_C1474/Csub_C1473 system separates from Ub/Rpn11 system. To me, results in this study indicate Csub_C1473 is a novel JAMM-type protease in Archaea (for example, archaeal JAMM-3 type in Fig.1) and Csub_C1474 is another type of Ub-like protein in Archaea (for example, another type of SAMP). You can only draw conclusions based on what the results show, and further claim that Csub_C1473 is archaeal Rpn11 and Csub_C1474 is archaeal ubiquitin is not right. Treating "Csub_C1473/Csub_C1474" system as first true archaeal Ub

modification system is inappropriate, especially when we consider archaeal SAMP system contains components with both sequence and function similarities to eukaryotic Ub, E1, E3, JAMM metalloprotease, AAA ATPase and 20S core particle (Humbard et al., 2010; Miranda et al., 2014; Miranda et al., 2011; Hepowit et al, 2016; Fu et al, 2017; Hepowit et al., 2012; Cao et al., 2017; Fu et al., 2016). In addition, this manuscript is mainly about "the structure of an archaeal JAMM (Csub_C1473)/Ub-like protein (Csub_C1474) complex" as the title shows. However, this complex adds little to our understanding of JAMM proteases in general, as both "eukaryotic JAMM/Ub complex" and "archaeal JAMM/Ub-like protein complex" structures have been well characterized in the cases of eukaryotic AMSH/AMSH-LP (Sato., 2008; Shrestha et al., 2014), Rpn11 (Worden et al., 2017) and archaeal JAMM-1 (Cao et al., 2017).

Comparative genomic studies have unearthed a number of features in Aigarchaeota (to which Caldiarchaeum belongs) and the Asgard archaea superphyla with properties akin to those found in eukaryotes. This includes the presence of CsRpn11, CsUb and ubiquitin conjugating proteins in Caldiarchaeum. Based on bioinformatic, phylogenetic and experimental evidence their features have been considered distinct and novel enough to designate these proteins as parts of a ubiquitin modifier system (Grau-Bové X et al., Mol Biol Evol 2015; Koonin, EV and Yutin, N, CSH Perspect Biol 2014; Williams, TA et al., Nature 2013; Hennell James, R et al., Nat Commun 2017; Eme, L et al., Nat Rev Microbiol 2017).

It lies in the nature of an ancestral system that it does not have the same complexity as an evolutionary more advanced system. Which is exactly why one can learn from such primordial model systems on the evolution of more complex features. Some of these features may be derived from proteins that originally acted in a different context. Caldiarchaeum, for example, does not have a 19S proteasome particle, yet CsRpn11 shares many features with eukaryotic 19S-embedded Rpn11. A related example are 19S-embedded Rpt-AAA-ATPases, whose archaeal homologs are found as separate entities that, like CsRpn11, can act in solution.

As is the case for the eukaryotic ubiquitination system, also the Caldiarchaeum ubiquitination system did not arise in a single step de novo. Obviously, some of its components' features can be traced to other members of their respective protein families, e.g. SAMPs and JAMM proteases

(Fig. 1). In that sense, these protein groups are certainly of great interest in an evolutionary context too. In addition, they are of course pivotal for archaeal physiology, as is evident from the numerous findings by Maupin-Furlow and colleagues, cited in our paper and by this reviewer.

1. Page6 "Remarkably, while prokaryotic JAMMs which cluster distant from Rpn11 are functionally associated with just a SAMP- or Ub- like protein and an E1-like homolog,"
Comment: Inaccurate. Not just a SAMP- or Ub-like protein, at least three Ub-like proteins reported (e.g., SAMP1, SAMP2 and SAMP3) and related with archaeal JAMMs.

With 'a SAMP- or Ub-like protein' we did not mean any number, but were referring to the type of protein as such. To clarify this, we have modified the sentence.

2. Page6 "Finally, *C. subterraneum* and Asgard archaea, whose JAMM sequences are most similar to eukaryotic Rpn11, contain in addition RING-type E3-homologs to complete the ubiquitination machinery." Comment: Don't agree. Need to modify this sentence. First, "RING-type E3-homologs" is inappropriate, because CSUB_C1477 is only homolog of RING finger protein, a component of Cullin- RING E3 ligase. Second, "to complete the ubiquitination machinery" is dangerous. For example, substrate receptor of E3 is missing. Although in vitro CsE1, CsE2 and CSUB_C1477 ubiquitinate non-physiological substrate (James et al., 2017), however, in vivo, other components (e.g. substrate receptor of E3) of ubiquitination machinery is probably needed to complete the ubiquitination of a substrate.

We have rephrased the sentence accordingly (p6).

3. Page6 "is found distant from classical Ub and more closely related to prokaryotic SAMPs like Moad and ThiS." Comment: Incorrect. No archaeal SAMPs like bacterial ThiS and

archaeal SAMP1 is bi-functional. First, no archaeal SAMPs like bacterial ThiS of thiamine biosynthesis. Archaea lack steps of bacterial ThiF-ThiS-ThiG-ThiH pathway of synthesizing the thiazole ring, instead archaea apply eukaryote-like Thi4 mechanism to mobilize sulfur to form the thiazole ring (Hwang et al., 2017). Second, archaeal SAMP1 could be isopeptide-linked to hundreds of protein substrates, like eukaryotic ubiquitin, however, bacterial MoaD doesn't have this feature. Archaeal SAMP1 is bi-functional, on one hand like eukaryotic ubiquitin for diverse protein substrates modification (Humbard et al., 2010; Dantuluri., 2016), on the other hand like bacterial MoaD participating in MoCo biosynthesis (Miranda et al., 2011; Cao et al., 2015).

We have altered the sentence on p6 and now compare the phylogenetic features of Urm1 to prokaryotic 'homologs' like MoaD and ThiS, which should be a more suitable term.

4. Page7 "CsUb is an extremely thermostable monomeric protein ($T_m > 95^\circ\text{C}$ in circular dichroism heat-denaturation studies, not shown)." Comment: "not shown" is not ok. Other places in the manuscript also have this "data not shown" problems.

The CD melting curve for CsUb is now shown in Supplementary Figure 1. There is no other 'data not shown' in the revised manuscript.

5. Page7 "In vivo, the protein is made as a precursor of 87 amino acids (Ub-pre), from which the last 9 residues are removed upon maturation." Comment: No data support. As you mentioned C. subterraneum could not be cultured so far, did you mean you co-express CsUb-pre (Csub_C1474) and CsRpn11 (Csub_C1473) in E. coli and find last 9 residues of CsUb-pre is removed by Csub_C1473 in E. coli as the in vivo assay? If so, show the data.

The altered sentence on p7 does not make any reference to in vivo studies anymore.

6. Page7 "CsUb also superimposes best with human Ub (DALI Z-Score 12.2), but is more distinct from so far characterized prokaryotic Ub-like proteins, such as H. volcanii Moad/SAMP1 (Z = 6.9) or ThiS/SAMP2 (Z = 5.4)." Comment: Don't agree. H. volcanii SAMP2 is not ThiS-like and reasons were mentioned above. SAMP2 is eukaryotic ubiquitin-like, based on both functional and structural similarities, for example the following reasons: First, SAMP2 could be isopeptide-linked to hundreds of protein substrates, like eukaryotic ubiquitin (Humbard et al., 2010). Second, SAMP2 isopeptide modified protein substrate (e.g., TBP2) could be degraded in eukaryotic-like ubiquitin-proteasome pathway, in which archaeal SAMP2 (resemble eukaryotic ubiquitin), JAMM2 (resemble eukaryotic Rpn11), AAA ATPase PAN2, Cdc48c (resemble eukaryotic Rpt) and 20S proteasomes (resemble eukaryotic 20S proteasomes) were participated (Fu et al., 2016). Third, SAMP2 could apply a structure conformation (when in complex with JAMM1) extremely similar to eukaryotic ubiquitin and is recognized by archaeal JAMM1 in a similar way as ubiquitin is recognized by eukaryotic JAMM (Cao et al., 2017). How does the structure of Ub compare to CsUb vs. PfSAMP2 in complex with JAMM1? Did you compare the similarity of Ub vs. CsUb to Ub vs. PfSAMP2 in complex with JAMM1? If they don't show significant difference, then the statement "CsUb also superimposes best with human Ub but is more distinct from SAMP1 or SAMP2" is incorrect.

Our statement is based on the DALI Z-scores, in which a higher Z-score implicates higher structural similarity. The requested DALI comparison of PfSAMP2 in complex with PfJAMM1 with HsUb yields a Z-Score of 5.4. As stated before, CsUb yields a much higher Z-Score of 12.2 when compared with HsUb. In contrast to this high structural similarity to eukaryotic HsUb, CsUb is rather dissimilar to archaeal SAMPs like HvSAMP1 (Z = 6.9) or HvSAMP2 (Z = 5.4). In line with the DALI scores, BLAST cannot detect any significant sequence similarity between PfSAMP2 and HsUb, whereas CsUb is highly similar in sequence to HsUb ($E = 3 \times 10^{-13}$) with an amino acid identity of 30%. These data further support our conclusions, and underscore the close evolutionary relationship of CsUb with eukaryotic ubiquitin. A table summarizing these DALI scores is now provided as Supplementary Table 1.

7. Page8 "The physiological relevance of these dimers is unclear, considering that for all determined JAMM structures the dimerization interfaces differ (Figure S2)." Comment: Inappropriate. You only show 4 examples in the FigureS2 to claim "all determined JAMM structures" (much more than 4 examples). In addition, 3 of 4 examples that you show are not physiological-relevant dimers, since they behave as monomer in solution and just show artificial dimer in crystal.

This is the point we are making. A dimer is seen in the crystal structure, while a monomer is detected in solution. We have clarified the sentence on p8 to point out that "the physiological relevance of these dimers is unclear, considering that the dimerization interfaces in the respective crystal structures differ."

8. Page8 "In structure superimpositions, CsRpn11 displays the highest similarity to S. cerevisiae Rpn11 (DALI Z-Score 21.1), but is less similar to so far determined structures of other prokaryotic JAMMs, such as P. furiosus JAMM-120 (Z = 15.1)." Comment: How about "CsRpn11 (Csub_C1473)" similarities to eukaryotic CSN5, BRCC36 and AMSH/AMSH-LP? It is necessary to show. For example, if Csub_C1473 is also very similar to AMSH or even more similar to AMSH, is that indicate Csub_C1473 is AMSH- like rather than Rpn11-like. I do have some concerns, especially when consider: First, Csub_C1473 is active independently without requirement to form JAMM+/JAMM- heterodimer or larger complex, which is more like AMSH rather than Rpn11. Second, Asgard archaea do have AMSH-related ESCRT homologs (Zaremba-Niedzwiedzka et al., 2017).

The requested DALI scores are as follows: CsRpn11-CSN5 (Z=19.2); CsRpn11-BRCC36 (Z=19.7); CsRpn11-AMSH (Z=15.3). These data demonstrate that the similarity of CsRpn11 to eukaryotic Rpn11 (Z=21.2) is the highest, the similarity to AMSH the lowest, which is in line with our statement and which supports our conclusions. We added the data on p8/9, the scores are presented in addition in a Supplementary Table 2.

9. Page 11 "Our results show that the Ub binding mode of AMSH, Rpn11 and CSN5 was already established in an archaeal ancestor" Comment: Inappropriate. Ub-like SAMP2 binding mode of archaeal JAMM1 is extremely similar to Ub binding mode of eukaryotic JAMM and has been shown in previous study (Cao et al., 2017).

We have changed the sentence accordingly (p11).

10. Page12 "In fact, homologs of eukaryotic DUBs have not been found yet in archaea." Comment: Incorrect. Did you mean non-JAMM metalloprotease type DUBs (for example, cysteine protease type DUBs)? if so, write it clearly.

The sentence has been modified (p12).

11. Page12 "Likewise, neither linearly linked human Ub (HsUb2), nor Lys48- or Lys63- isopeptide linked HsUb2 were suitable substrates (Figure 4E)." Comment: This evidence appears opposing your statement that Csub_C1474 is CsUb-pre and Csub_C1473 is CsRpn11. It supports Csub_C1474 is another type/group of archaeal Ub-like proteins and Csub_C1473 is Csub_C1474-specific protease. In contrast, archaeal JAMM1 is more like Rpn11 in its ability to cleave true Ub-Ub dimers (Lys48 or Lys63 linkage) (Cao et al., 2017), compared to the archaeal Csub_C1473 ("CsRpn11").

If the features of a substrate sequence happen to allow an interaction with the protease (see interaction sites in Fig. 3), it is conceivable that also non-native substrates are being cleaved. Functionally, mechanistically and evolutionary, however, there is no reason why CsRpn11 would have to cleave eukaryotic ubiquitin. Nor do archaeal JAMM proteases in general have a more relaxed substrate specificity. CsJAMM, as we now show with new data in Supplementary Fig. 8A, does not cleave eukaryotic ubiquitin dimers either, the interaction sites presumably do not match sufficiently. Nor can it cleave CsUb. It can only cleave the two

CsSAMP precursors Csub_C0702 and Csub_C1603 (Supplementary Fig. 8A). As we have pointed out (p12), the parallel presence of two systems in the Caldiarchaeum cell (CsUb/CsRpn11 and CsSAMPs/CsJAMM) may have resulted in a rather stringent substrate specificity of the two proteases to avoid cross-reactivity. The new CsJAMM results, which support our hypothesis, have been added on p12.

12. Page 12-13 "At 45°C, truncated CsRpn11 processed CsUb-pre with a KM of 24.2 μM and a kcat of 3.5/min and bound the reaction product with a KD of 14.6 μM. With a KM of 31.6 μM and a kcat of 4.0/min, the full-length version displayed similar reaction parameters, indicating that the CsRpn11 C- terminal helices are indeed not critically required for catalysis. " Comment: Why perform at 45 °C? Is it optimized temperature for CsRpn11 activity or close to physiological temperature that C. subterraneum grow/live? If neither of them, it is hard to know whether "C- terminal helices" impact catalysis or not. For example, if 45°C is far away from optimized temperature of CsRpn11 activity, the difference of kinetics behavior of CsRpn11 (full length) and CsRpn11 (without C- terminal helices) may not be significant, but if tested at optimized temperature, the much higher difference may occur.

CsRpn11 performs very well at 45°C, but can be assayed up to 60°C, which would be in the physiological growth range of Caldiarchaeum. The reaction kinetics of CsRpn11 and CsRpn11^{Δ149-202} at 60°C are now shown in Supplementary Fig. 9. As expected, both proteins are similarly active, confirming that the C-terminal helices, which are absent in other JAMM proteases, do not influence CsRpn11 activity, neither at 45°C nor at 60°C.

13. Page14 "Though to our knowledge metal ion specificity has not been tested for other JAMM proteases," Comment: Incorrect. Metal ion specificity was tested in HvJAMM1 before (Hepowit et al., 2012).

We have missed this, presumably because only Zn²⁺ was reported to result in an active JAMM protease. The statement on p14 has been modified accordingly.

14. Page14 paragraph about "8-TQ inhibit CsRpn11 (Csub_C1473)"; Page15 "The susceptibility of CsRpn11, in particular for the highly selective 8-QT, thus underscores the close relationship between archaeal and eukaryotic Rpn11." Comment: Further experiments needed. Please aware other eukaryotic JAMMs are also sensitive to 8-QT/capzimin (8-QT derivative) (Li et al., 2017), only based on Csub_C1473 susceptibility to 8-QT without comparison, it could not indicate Csub_C1473 close relationship to Rpn11. Besides, archaeal JAMM-1 may also be sensitive to 8-QT. Please consider the following questions and possibilities. Whether Rpn11-specific inhibitor 8-QT also inhibits archaeal JAMM-1 type protease (e.g., PfJAMM1 or HvJAMM1)? If 8-QT has stronger inhibition effect to archaeal JAMM-1 compared to "Csub_C1473 (CsRpn11)", is that indicating archaeal JAMM-1 rather than "Csub_C1473 (CsRpn11)" is close to Rpn11? How about Csub_C1473 sensitivity to eukaryotic CSN5-specific inhibitors (Schlierf et al., 2016; Altmann et al., 2017)? If "CsRpn11 (Csub_C1473)" is more sensitive to CSN5-specific inhibitor than Rpn11-specific inhibitor, is that indicating Csub_C1473 is CSN5-like rather than Rpn11-like?

The inhibitors thiolutin and 8-TQ were the topic of recent high-profile publications, as cited in our paper, with the authors' attention focusing on Rpn11. As only eukaryotic proteins were analysed so far, we were curious to see possible effects on prokaryotic proteases and tested CsRpn11, as it is the subject of this paper. It was not our intention to use inhibitor binding as a way to establish phylogenetic relationships. We agree with the reviewer that this is not feasible, and testing various proteins with various inhibitors would not provide useful information in that regard.

The binding of thiolutin and 8-TQ to other archaeal JAMM proteases is of course a distinct possibility. We followed the reviewer's suggestion and exposed CsJAMM to the inhibitors.

Indeed, both thiolutin and 8-TQ affect its proteolytic activity, albeit the effects on CsJAMM are significantly weaker than the ones seen on CsRpn11 (Supplementary Fig. 8B, Fig. 5C). While this is in line with the higher similarity of CsRpn11 to eukaryotic Rpn11 (see point 8), it is equally possible that other reasons account for this result. We have added these findings on p15 and p25 and modified the corresponding paragraph.

15. Page15 "This procedure generated mono- and di-ubiquitinated TEV proteases (CsUb-TEV and CsUb2-TEV)." Comment: Does di-ubiquitinated TEV mean two CsUb as a chain linked to TEV or two mono-CsUb linked to TEV?

Two mono-CsUbs are linked to TEV. A poly-CsUb chain is chemically not possible due to the use of a CsUb^{C33A} mutant devoid of internal cysteines (p27). We have clarified the description on p16.

16. Page17 "Conclusions" paragraph Comment: Too many "discussion" exist in this paragraph.

The heading has been altered.

17. Page 17 "While Ub conjugation was long thought to be restricted to eukaryotes and to have evolved from distant prokaryotic homologs, such as the ThiS or MoaD conjugation system." Comment: Incorrect. ThiS or MoaD systems are not conjugation systems, but SAMP and ubiquitin systems are. ThiS and MoaD system are systems of thiamine biosynthesis and MoCo biosynthesis, respectively.

We have changed the sentence accordingly.

18. Page18 "bioinformatic discovery of complete Ub modification systems in C.

subterraneum, and more recently in Asgard archaea, suggests that the fundamental function of this system, protein tagging, was already established in archaea. The first true archaeal Ub modification systems, however, co-existed side by side with the universal archaeal SAMP/JAMM system," Comment: This statement indicates "archaeal CsUb-pre (Csub_C1474)/CsRpn11 (Csub_C1473)" and "archaeal SAMP/JAMM system" are two different systems, which I don't agree. To me, the results of this study indicate Csub_C1474 is another type of ubiquitin-like protein in archaea (for example SAMP4), besides SAMP1, SAMP2 and SAMP3. Csub_C1473 is another type (JAMM-3 type) of JAMM proteases in archaea, besides JAMM1 and JAMM2. This statement indicates "CsUb-pre (Csub_C1474)/CsRpn11 (Csub_C1473)" system is first true archaeal Ub modification systems, which I don't agree. Asgard archaea and C. subterraneum have eukaryotic E3-like component (RING-finger protein) is a main point that the authors think "CsUb-pre (Csub_C1474)" system is first true archaeal Ub modification systems. However, SAMP system also has eukaryotic E3-like component, for example, archaeal MsrA (Fu et al., 2017). Archaeal MsrA is the homolog of eukaryotic CRBN of E3 ubiquitin ligase CUL4-RBX1-DDB1-CRBN (Fischer et al., 2014), and archaeal MsrA participates in Ub-like SAMP conjugation to substrates (Fu et al., 2017), function also similar to eukaryotic E3.

It is a very interesting finding that the methionine sulfoxide reductase MsrA could help to direct SAMP to proteins. We are not sure though if MsrA could serve as an example for a eukaryotic E3-like component, a designation which Fu et al. actually do not make either. MsrA does not have E3 hallmark RING, HECT or RBR domains, there is no E2-like protein it could interact with, and there is no evidence that MsrA binds substrates (Fu, X et al., MBio 2017).

Thus, while SAMP systems may have evolved their own ways to target proteins, the ubiquitination machineries of Asgard archaea or Caldiarchaeum would at present seem to be model systems of choice to study the evolution of eukaryotic ubiquitination. The reasons

leading to this view have been given already in the introduction to our response above, and exemplary references were cited. In addition we refer to our own bioinformatic and experimental results, all of which are supportive of this view.

19. Page19 "one can ask how an archaeal system could possibly operate with just a few components, with for example only two small E3 RING finger proteins in C.

subterraneum, compared to hundreds of sophisticated E3 enzymes in eukaryotes"

Comment: Misleading, need to modify this statement. First, RING finger proteins in eukaryotes are also limited (e.g., Rbx1, Rbx2), instead of hundreds. Substrate receptors of E3 in eukaryotes are "hundreds" or even more. A lot of substrate receptors of E3 in archaea including C. subterraneum may also exist.

Second, the claim "one can ask how an archaeal system could possibly operate with just a few components" is based on limited in vitro evidence, since the organism C. subterraneum can not be cultivated so far.

We do not claim at all that the Caldiarchaeum ubiquitin system operates with just a few components. On the contrary, we unbiasedly discuss how substrates might be selected. As we state in the last sentence of the discussion, we consider this question to be of central importance. In humans, there are approx. 600 RING E3s, 28 HECT E3s and 14 RBR E3s (Buetow, L and Huang, DT, Nat Rev Mol Cell Biol 2016). Given that the two small Caldiarchaeum RING finger proteins do not seem to have substrate binding domains, unlike the eukaryotic E3s, we wonder how in this case substrate recognition works. The answer to that question may very well entail the existence of other, as yet unidentified components.

20. Figure 1 and Figure 1 legends Comment: Need some modification based on reasons mentioned in previous comments. For one example, "Archaeal Moad" and "Archaeal This" are inappropriate and misleading.

We refer to the landmark papers by Burroughs, AM et al (Front Biosci 2012) and Makarova, KS and Koonin, EV (Archaea 2010), which bioinformatically delineate 5-stranded beta grasp proteins, to which ubiquitin and SAMPs belong, into ThiS and MoaD superfamilies. This classification is based on structural and sequence similarities, irrespective of potential functions, and includes archaeal proteins in both clades. Nonetheless, we renamed the respective clusters “archaeal MoaD/SAMP1” and “archaeal ThiS-like/SAMP2”.

21. Figure 4 Comment: The Mass results appear wrong. Figure 4A show CsUb-pre 12.462kDa and CsUb 11.344 kDa. However, based on my calculation Csub_C1474 (CsUb-pre) of 87 amino acid MW: 9.11 kDa; CsUb of 78 amino acid MW: 8.15 kDa. If you have some specially treatment to "Csub_C1474 (CsUb-pre)", please write clearly. In addition, no standard (e.g., CsUb or CsUb-pre with known amount) was loaded on the same gel, how could you quantify gel bands of CsUb or CsUb-pre for kinetics calculation?

As described in Methods (p21), CsUb-pre was produced with an N-terminal His₆-tag and a TEV protease cleavage site. These additional sequences account for the extra mass. In our assays, we observed no difference between tagged and untagged CsUb-pre. The figure legend has been modified for clarity.

For the kinetics calculation, it goes without saying that a standard was loaded on the same gel, which can now be seen in Supplementary Fig. 7A and B, where full gels are shown. The description in Methods has been modified to make this point clearer (p26).

22. Figure 6 Comment: Control needed (e.g, "EDTA-treated Rpn11 with substrate") to make sure the "CsUb" band is due to "Csub_C1473 (CsRpn11)" cleavage of substrate. As I concern: during incubation, a little portion of substrate may be reduced and CsUb band occur.

We replaced Figure 6 and have added a control with the chelator. As expected, EDTA-treated Rpn11 does not cleave CsUb, nor was there inadvertent substrate reduction during the incubation.

Reviewer #2 (Remarks to the Author):

The authors provide a biochemical and structural analysis of an archeal JAMM-type deubiquitinating enzyme (csRpn11), which is remarkably similar to its eukaryotic Rpn11 counterpart. The different csRPN1 structures described in this manuscript - with and without bound archeal ubiquitin - are representatives of the different catalytic steps of the DUB reaction. This manuscript is very interesting for the evolution of the ubiquitin pathway.

The work described is solid and well-executed; the representation and discussion of the results are clear and leave little to be desired. However, there are a few minor points that could be improved on.

1. page 7, lower part: it is not correct to claim that eukaryotic ubiquitin has 'equivalent Lys residues' compared to csUb. As the authors correctly state later, the lysines are found at different positions, which is hardly reconcilable with their 'equivalence'.

Indeed, the word 'equivalent' was misleading in this context, we removed it (p7).

2. page 8, bottom line: what is meant by the term 'circular JAMM protease core' ? I can't see anything circular here.

The seven beta strands assemble into a circular barrel, which admittedly is hard to see in Fig. 3B due to the angle at which the protein is shown. We have omitted the word 'circular' and changed the description of the JAMM protease core slightly (p9).

3. page 9ff, in connection to figure 3: The authors describe several structural features (helices, insert regions), which are not indicated in the structure. In general, I think that structure 3B should be rendered bigger and should contain labels for the insert regions and other features mentioned in the text.

We very much appreciate this suggestion, and agree that the now enlarged Figure 3B, together with the labels, provides a much clearer picture of the proteins' features.

4. page 11, top, explains the contribution of particular residues by referring to the yeast ScRpn11 sequence. However, the alignment figure 3A shows only archeal and human Rpn11 sequences, making it very hard to appreciate discussions of "Gly-66 in ScRpn11". The authors should either use the residue numbering of human Rpn11 or alternatively include the yeast sequence in figure 3A.

*Figure 3A now shows the *S. cerevisiae* Rpn11 sequence instead of the human Rpn11 sequence, and Gly77 is highlighted. The location of Gly77 is also indicated in Supplementary Fig. 5.*

5. page 13, the authors write "..the linear substrate Ub-TAMRA.." I don't see why Ub-TAMRA should be considered a "linear substrate". The manuscript does not describe the source of the the Ub-TAMRA used by the authors, but the commercially available Ub-TAMRA has the Ub-cterminus bound to the epsilon-amino group of a lysine residue (which in turn is coupled to TAMRA). Thus, Ub-TAMRA emulates an isopeptide linkage rather than the peptide linkage found in linear chains. The authors should clarify this.

We have corrected the sentence describing Ub-TAMRA accordingly (p13).

6. Figure 4e: I wonder why in this gel, the linear HsUb2 runs at a much higher MW than the K48- and K63-linked diUb species. In our hands, all three human diUb species run (almost) at the same height. Since there is no MW ladder provided, it is not clear how big the apparent MW difference really is. Also, I did not find a source for the used diUb species in the materials section. This should be rectified.

We added the source for the diUb proteins in the Method section (p25) and also added molecular weight information for all gels. As can be better seen in the full gel (Supplementary Fig. 7C), also in our hands the proteins run at almost the same height.

REVIEWERS' COMMENTS:

Reviewer #2 (Remarks to the Author):

The authors have done a good job addressing my complaints. There are no further requests from my side.